# Learning What Matters: Toward Causally-Grounded Foundations for Contextual Time-Series Forecasting

## Abstract

Time-series evolution is driven by intrinsic dynamics and heterogeneous contextual factors whose relevance can be spurious or unstable across settings, which limits correlation-driven designs that impose fixed structures on contextual inputs. **We propose a causally grounded foundation model** for context-aware forecasting that augments a generative pre-trained transformer with a context-aware attention module to adaptively integrate external signals. From an information-theoretic perspective, a **query-modulation** mechanism conditions temporal queries on global context summaries, reducing redundancy and sharpening focus, while a neural structural-equation component injects inductive bias toward genuinely influential features. To align attribution with behavior, **counterfactual perturbations** enforce consistency between structural importance and predictive responses; an entropy-based regularizer further encourages sparse, interpretable attributions. The overall design targets accuracy, robustness to distributional shifts, and explanatory clarity in the presence of noisy or weakly relevant context. Extensive experiments on diverse real-world benchmarks demonstrate consistent gains over strong baselines in point forecasting and calibration, alongside clearer and more stable explanations of context contributions.

## 1 Introduction

Time series forecasting traditionally aims to predict future values by modeling the temporal dependencies of the target variable based on its historical patterns (Lim & Zohren, 2021; Torres et al., 2021; Chang et al., 2023). This paradigm underlies a wide range of classical and modern approaches, from statistical models Ahmed et al. (2010); Bontempi et al. (2012); Montgomery et al. (2015) to deep learning architectures Shen et al. (2020); Benidis et al. (2022); Zhang & Yan (2023); Jin et al. (2024); Das et al. (2024); Woo et al. (2024); Wang et al. (2024a). However, in many real-world scenarios, future values are shaped not only by historical trajectories, but also by surrounding factors that influence or perturb the system (Williams et al., 2024; Chen et al., 2024a; Jia et al., 2024). A spike in electricity demand may reflect not a natural trend, but a sudden temperature drop; a collapse in stock price may be triggered not by technical signals, but by a regulatory headline. These changes arise not from the sequence itself, but from external factors that act upon it. Recognizing this subtle yet critical distinction shifts the focus of forecasting beyond mere continuation, toward uncovering and modeling the mechanisms that truly drive change. Accordingly, how to effectively and accurately leverage contextual information to enhance forecasting accuracy has become a central research question (Chattopadhyay et al., 2024; Wang et al., 2024b; Liu et al., 2024a; Yang, 2025; Wang et al., 2025).

**Fundamental issues in context-aware forecasting.** Organizing and integrating contextual information into time series forecasting presents several fundamental challenges. **(1) Heterogeneous context modalities.** Contextual variables span diverse modalities such as temporal encodings, structured covariates and textual descriptions, with substantial variation in format and semantics across datasets (Chattopadhyay et al., 2024; Wang et al., 2024b). This heterogeneity hinders the construction of a unified and extensible input space, especially in settings that demand cross-domain transferability or large-scale pretraining. **(2) Contextual noise and spurious correlation.** While some contextual features are informative, others reflect spurious correlations or unstable patterns

that shift across domains (Ye et al., 2024; Meng et al., 2023; Cao et al., 2023a). Models that absorb all signals indiscriminately are prone to shortcut learning and performance degradation under distribution shifts. While methods like SAMformer (Ilbert et al., 2024) enhance robustness to spikes using sharpness-aware optimization, they still rely on architectural smoothing instead of reasoning about the causes. These issues call for a principled approach to identify and prioritize causal factors that genuinely influence the prediction target. **(3) Generalization across contexts.** Real-world deployments increasingly demand general-purpose forecasters capable of adapting to diverse context structures (Chattopadhyay et al., 2024; Yang, 2025; Wang et al., 2025). Meeting this need requires an architecture that supports unified input representation and adaptive context selection, enabling scalable application of context-aware foundation models.

**Limitation of current approaches.** Recent advances have taken promising steps toward context-aware forecasting (Liu et al., 2024a; Yang, 2025; Wang et al., 2025). **(1) Rigid context representation.** Approaches like Context is Key (Williams et al., 2024) embed structured contextual signals into the modeling process, strengthening the semantic expressiveness of time series representations. However, by unifying contexts such as covariates and temporal features into fixed textual descriptions, these methods constrain representational flexibility and limit scalability across diverse tasks. **(2) Correlation-driven integration.** Another line of work, such as Context Matters (Chattopadhyay et al., 2024) and TimXer (Wang et al., 2024b), adopts an attention-based correlation-driven paradigm to integrate context. While capable of capturing potentially relevant factors, these models lack causal reasoning and are easily misled by spurious correlations, limiting their generalizability. **(3) Shallow semantic alignment.** More recent efforts, such as Time-LLM (Jin et al., 2023), incorporate contextual modeling into large language models, leveraging their flexibility in handling diverse inputs through natural language representations. However, these methods often rely on similarity-based strategies, such as embedding matching or shallow attention, which limit the ability to capture structural relationships and temporal dynamics crucial for accurate forecasting. Despite these promising developments, there remains a pressing need for a new generation of foundation models that can accurately identify and leverage truly causal contextual signals, enabling robust generalization across different datasets.

To address the aforementioned challenges, we propose a causally grounded foundation model for context-aware time series forecasting. Our approach is built upon a generative pre-trained transformer, augmented with a lightweight context-aware attention module designed to improve adaptability to diverse and complex contextual structures. From an information-theoretic perspective, we introduce a modulation mechanism that dynamically conditions temporal queries on the global semantics of contextual features, allowing the model to suppress redundant signals while retaining informative cues. To move beyond correlation-based attention, we introduce a neural structural equation module that encodes a global causal inductive bias, offering a principled estimation of each context variable's structural influence. This bias is directly injected into the attention computation, transforming attention from a similarity-driven heuristic into a causally aligned inference mechanism. Crucially, we close the causal loop through a counterfactual validation process: by systematically perturbing high-salience features during training, we require the model to align its predictions with interventional outcomes. This behavioral signal enforces that attribution is not only plausible but functionally necessary. Finally, an entropy-based regularizer further promotes sparse, interpretable structure in the learned causal map. Together, these components work toward disentangling causal context, reinforcing attribution reliability, and supporting generalization across diverse environments.

In summary, our contribution can be summarized in three folds:

- We propose a causally grounded foundation model for context-aware time series forecasting, which integrates generative pre-trained transformers with a context-aware architecture to support generalization across diverse and dynamic forecasting scenarios.

- We develop a causal context modeling framework that jointly incorporates a neural structural equation module and a counterfactual intervention mechanism. This design injects causal inductive bias and encourages the model to focus on causally relevant context signals by aligning attribution with interventional outcomes.

- We conduct extensive experiments on real-world benchmarks, demonstrating substantial improvements in forecasting accuracy. Our model also provides interpretable context selection that highlights which contextual features contribute to predictive performance.

## 2 RELATED WORK

### 2.1 CONTEXT-ENHANCED TRANSFORMER MODELS FOR TIME SERIES FORECASTING

Incorporating auxiliary signals has long been a key strategy in time series forecasting. Classical models like ARIMAX Contreras et al. (2003) and SARIMAX (Dubey et al., 2021) include exogenous variables via autoregressive structures, while deep learning models such as DeepAR (Salinas et al., 2020), TFT (Lim et al., 2021), and NBEATSx (Olivares et al., 2023) integrate covariates through concatenation or attention. Recent methods like TiDE (Das et al., 2023), TimeXer (Wang et al., 2024b), and TSMixer (Chen et al., 2023) enhance flexibility through patching or metadata encoding but lack mechanisms to filter spurious signals. Extensions to multimodal and textual context (Xu et al., 2024; Hong et al., 2025) have led to benchmarks like CONTEXT IS KEY (Williams et al., 2024) and models such as ContextMatters (Chattopadhyay et al., 2024), ChatTime (Wang et al., 2025), and CALF (Liu et al., 2025b), though most overlook causal relevance. In parallel, Transformer-based models have become central to time series forecasting due to their scalability and ability to capture long-range dependencies (Zhou et al., 2021; Liu et al., 2022; Chen et al., 2024b; Zhang & Yan, 2023). Architectures like Autoformer (Wu et al., 2021), PatchTST (Nie et al., 2022), and iTransformer (Liu et al., 2023) adopt sparse attention and patching to improve efficiency. Some recent methods, such as Time-LLM (Jin et al., 2023), GPT4TS (Zhou et al., 2023), and UniTime (Liu et al., 2024b), explore prompt-based or instruction-driven approaches, while others focus on large-scale pretraining over raw series (Ansari et al., 2024; Woo et al., 2024; Garza et al., 2023; Rasul et al., 2023; Das et al., 2024). Despite their strengths, most of these models treat time series in isolation and underutilize contextual signals critical for robust generalization.

### 2.2 CAUSAL MODELING IN TEMPORAL SETTINGS

Temporal causal modeling began in epidemiology and is often based on G-computation, marginal structural models (MSMs), and structural nested models Robins (1986); Robins et al. (2000); Robins & Hernan (2008). Early methods relied on linear assumptions and struggled to capture complex temporal patterns. Later works improved expressiveness by using Bayesian nonparametrics (Xu et al., 2016; Soleimani et al., 2017; Schulam & Saria, 2017) and deep models such as RMSNs (Lim, 2018) and G-Net (Li et al., 2021), which apply RNN-based architectures to forecast treatment outcomes. Recent approaches aim to learn representations that are both predictive and invariant to treatment (Ganin et al., 2016; Tzeng et al., 2015). Examples include CRN (Bica et al., 2020) and Transformer-based models (Melnychuk et al., 2022; Vaswani et al., 2017), which often use joint losses or domain confusion. Some studies extend counterfactual reasoning to irregular time series (Seedat et al., 2022; Cao et al., 2023a), point processes (Zhang et al., 2022), and spatiotemporal graphs (Jiang et al., 2023; Kidger et al., 2020). Although many methods focus on balancing, recent work shows that expressiveness plays a more important role in robust performance Meng et al. (2023). To handle unmeasured confounders, some models use proxies or frequency-based decomposition Cao et al. (2023a), but these often require strong assumptions and may not generalize well.

## 3 METHOD

**Notation.** Let $L$ be the history length, $H$ the forecast horizon, $N$ the number of context channels, $d$ the latent dimension, and $D_y$ the target dimension. We denote the observed target history by $\mathbf{T} \in \mathbb{R}^{L \times D_y}$ with rows $\mathbf{t}_\ell^\top \in \mathbb{R}^{D_y}$, and the future to be predicted by $\mathbf{t}_{L+1:L+H} \in \mathbb{R}^{H \times D_y}$. The unified context matrix is $\mathbf{C} \in \mathbb{R}^{L \times N}$, where each column is one context channel aligned to time. Patch-embedded context tokens are $\mathbf{Z}_c \in \mathbb{R}^{N \times d}$, and the attention queries/keys/values are $\mathbf{Q} \in \mathbb{R}^{L \times d}$, $\mathbf{K}, \mathbf{V} \in \mathbb{R}^{N \times d}$.

### 3.1 PROBLEM FORMULATION

We consider the task of context-aware time series forecasting—focusing on predictive modeling rather than identification of causal effects—where the objective is to model the conditional distribution of future observations given historical data and auxiliary contextual signals. Formally, given

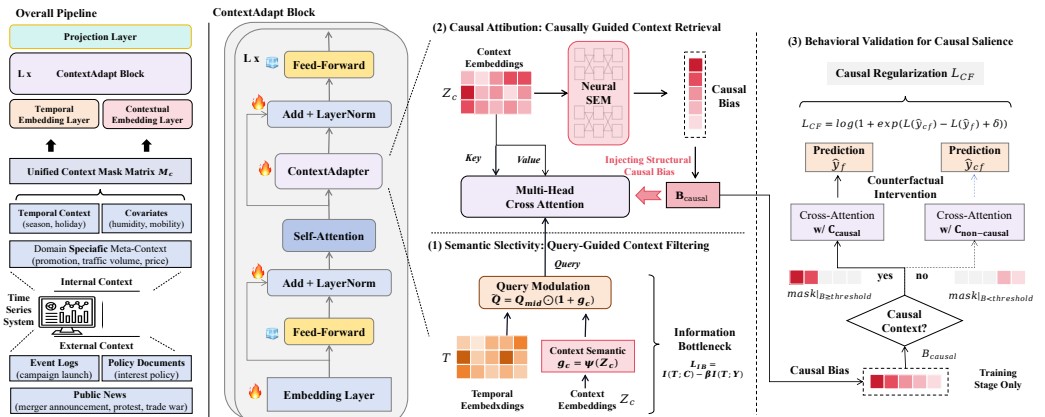

Figure 1: Overview of our framework. Contextual inputs are processed via a unified encoder, followed by a three-stage causal modeling pipeline: (1) **Semantic Selectivity** modulates queries to reduce spurious dependencies, (2) **Causal Attribution** estimates structural relevance to guide cross-attention, and (3) **Behavioral Validation** enforces alignment through counterfactual interventions.

history $\mathbf{T} \in \mathbb{R}^{L \times D_y}$ and context $\mathbf{C} \in \mathbb{R}^{L \times N}$, we learn

$$p(\mathbf{t}_{L+1:L+H} \mid \mathbf{T}, \mathbf{C}), \qquad \mathbf{t}_{L+1:L+H} \in \mathbb{R}^{H \times D_y}. \tag{1}$$

We adopt a system-centric view of context: *internal* context (intrinsic covariates and temporal features) and *external* context (exogenous information such as event logs, text, or policies). We distinguish between *internal context*, which comprises variables intrinsic to the time series system, and *external context*, which originates outside the system but may influence its behavior. Internal context includes temporal features (e.g., year, season, weekday, holiday), co-evolving covariates, and domain-specific metadata (e.g., promotions in sales forecasting, pricing in electricity markets, or traffic volume in transportation systems). External context includes structured or unstructured information such as event logs, textual descriptions, or policy documents that affect the target sequence through external mechanisms. A complete specification of all contextual inputs is provided in Appendix C.

Following recent advances in time series modeling, we adopt a generative pre-trained transformer (GPT) architecture as the backbone. While this architecture effectively models the intrinsic dynamics of the target sequence $\mathbf{T}$, it does not naturally account for auxiliary contextual signals $\mathbf{C}$ that often influence future outcomes in real-world scenarios. To address this limitation, we introduce a dedicated context-aware cross-attention module that conditions the representation of $\mathbf{T}$ on information retrieved from $\mathbf{C}$. We aim to replace correlation-based integration with a structured, causally guided mechanism that enables selective context use and consistent reasoning about its impact. The overall architecture is illustrated in Figure 1.

## 3.2 UNIFIED CONTEXT INTEGRATION AND MASKED ENCODING

To standardize heterogeneous contextual sources, we construct a unified matrix $\mathbf{C} \in \mathbb{R}^{L \times N}$ and a binary mask $\mathbf{M_c} \in \{0, 1\}^{L \times N}$ that activates only available channels per instance. The masked context is embedded as

$$\mathbf{Z}_c = \text{PatchEmb}(\mathbf{M_c} \odot \mathbf{C}), \qquad \mathbf{Z}_c \in \mathbb{R}^{N \times d}, \tag{2}$$

where $\odot$ denotes element-wise multiplication. This formulation provides a modular interface for encoding heterogeneous context into a shared latent space.

## 3.3 STRUCTURALLY GROUNDED ATTENTION FOR CAUSAL CONTEXT INTEGRATION

We reinterpret cross-attention as a structured interface for context integration that is *aligned with causal considerations*, moving beyond purely correlation-based retrieval. To support robust generalization, we design a controllable mechanism that selects contextual signals by their predictive

utility and structural relevance: (1) **semantic selectivity**, which modulates queries to suppress redundant or spurious dependencies; (2) **structural attribution**, which injects a neural structural equation module to produce a nonnegative, globally learned bias over context channels; and (3) **behavioral validation**, which applies counterfactual masking of high-bias channels to enforce reliance consistent with the declared importance. Together, these components convert attention from a passive retrieval heuristic into an active mechanism for selective, interpretable, and causally *aligned* context use (without claiming identifiability of causal effects).

**Context-Aware Query Modulation.** Motivated by the information bottleneck (IB) principle, we aim for query representations that preserve task-relevant information from the context while suppressing nuisance dependencies. Let $T$ denote the (random) query representation induced by the target stream, $C$ the context, and $Y$ the future targets. We consider the conceptual objective

$$\mathcal{L}_{\text{IB}} = I(T;C) - \beta\, I(T;Y), \tag{3}$$

with $\beta > 0$. Direct optimization of (3) is intractable, so we adopt a soft, parametric implementation in the query space.

Given the intermediate queries $\mathbf{Q}_{\text{mid}} \in \mathbb{R}^{L \times d}$ and the context tokens $\mathbf{Z}_c \in \mathbb{R}^{N \times d}$, we compute a global context summary $\mathbf{g}_c = f(\mathbf{Z}_c) \in \mathbb{R}^d$ and modulate the queries as

$$\tilde{\mathbf{Q}} = \mathbf{Q}_{\text{mid}} \odot \left( \mathbf{1}_{L \times d} + \mathbf{1}_L\, \mathbf{g}_c^\top \right) \in \mathbb{R}^{L \times d}, \tag{4}$$

where $\mathbf{1}_L \in \mathbb{R}^L$, $\mathbf{1}_{L \times d} \in \mathbb{R}^{L \times d}$ are all-ones tensors, and $\odot$ denotes element-wise multiplication. Equation (4) broadcasts $\mathbf{g}_c$ across the $L$ time positions, steering the queries toward globally salient context semantics and thereby encouraging compact yet discriminative retrieval.

**Causal Inductive Bias via Neural SEM.** Given the context tokens $\mathbf{Z}_c \in \mathbb{R}^{N \times d}$, we introduce a neural structural equation module (SEM) that outputs a nonnegative, channel-wise salience vector

$$\mathbf{B}_{\text{causal}} = \psi_{\text{SEM}}(\mathbf{Z}_c) \in \mathbb{R}_{\geq 0}^N, \tag{5}$$

parameterized with a nonnegative head (e.g., Softplus or Sigmoid×scale) for interpretability and comparability across channels. We promote decisiveness via an entropy regularizer

$$\mathcal{L}_{\text{ent}} = \mathbb{H}(\text{Softmax}(\mathbf{B}_{\text{causal}})) = -\sum_{i=1}^N \pi_i \log \pi_i, \quad \pi_i = \frac{\exp(B_{\text{causal},i})}{\sum_{j=1}^N \exp(B_{\text{causal},j})}. \tag{6}$$

The salience $\mathbf{B}_{\text{causal}}$ serves as a *global* structural prior (aligned with causal considerations but without claiming identifiability).

We inject the global bias into cross-attention to guide retrieval toward structurally salient channels:

$$\text{Attn}(\mathbf{Q}, \mathbf{K}, \mathbf{V}) = \text{Softmax}\left( \frac{\mathbf{Q}\mathbf{K}^\top}{\sqrt{d}} + \mathbf{1}_L\, \mathbf{B}_{\text{causal}}^\top \right) \mathbf{V}, \quad \mathbf{Q} \in \mathbb{R}^{L \times d},\ \mathbf{K}, \mathbf{V} \in \mathbb{R}^{N \times d}, \tag{7}$$

where $\mathbf{1}_L \mathbf{B}_{\text{causal}}^\top \in \mathbb{R}^{L \times N}$ broadcasts the channel bias to all $L$ query positions. This combines instance-specific relevance (query–key similarity) with a globally learned structural prior.

**Behavioral Validation via Counterfactual Masking.** Even with a global structural prior, a central question remains: does the model *actually rely* on the channels it declares important? We therefore construct a counterfactual environment that removes access to the most salient context channels and quantify the induced behavioral change. Let

$$S_k = \text{TopK}(\mathbf{B}_{\text{causal}})$$

be the indices of the $k$ largest entries in $\mathbf{B}_{\text{causal}}$ (nonnegative by construction). Let $\mathcal{P}_{\overline{S_k}}(\cdot)$ be a column-wise masking operator that zeros (or replaces by the per-channel mean) the channels indexed by $S_k$ while leaving other channels intact. We obtain the factual prediction $\hat{\mathbf{t}}_f \in \mathbb{R}^{H \times D_y}$ using the original context $\mathbf{C}$, and the counterfactual prediction $\hat{\mathbf{t}}_{cf} \in \mathbb{R}^{H \times D_y}$ by replacing $\mathbf{C}$ with $\mathcal{P}_{\overline{S_k}}(\mathbf{C})$ in the same forward path (the bias $\mathbf{B}_{\text{causal}}$ is *not* altered during masking). We then define a contrastive loss

$$\mathcal{L}_{\text{CF}} = \mathbb{E}\left[ \log\left(1 + \exp\left(\ell_{\text{pred}}(\hat{\mathbf{t}}_{cf}, \mathbf{t}_{L+1:L+H}) - \ell_{\text{pred}}(\hat{\mathbf{t}}_f, \mathbf{t}_{L+1:L+H}) + \delta\right)\right) \right], \tag{8}$$

where $\ell_{\text{pred}}$ is the pointwise prediction loss (e.g., MSE over $H \times D_y$) and $\delta > 0$ is a margin. This "behavioral validation" encourages the model to truly rely on channels it declares important.

Together, $\mathcal{L}_{\text{CF}}$ and $\mathcal{L}_{\text{ent}}$ induce a structural filter $S_k = \text{TopK}(\mathbf{B}_{\text{causal}})$. We write $C_{S_k} \subseteq C$ for the sub-matrix obtained by selecting the columns of $C$ indexed by $S_k$; this notation is used below to analyze sufficiency.

**Proposition 1** (Approximate Minimal Sufficiency). *We write $C_{S_k} \subseteq C$ for the sub-matrix of $C$ obtained by selecting the columns indexed by $S_k$, where $S_k = \text{TopK}(\mathbf{B}_{\text{causal}})$. The bias vector $\mathbf{B}_{\text{causal}}$ is a deterministic function of $C$ via the SEM, i.e., $\mathbf{B}_{\text{causal}} = \psi_{\text{SEM}}(Z_c(C))$, and it is not a probability distribution. Under joint optimization of $\mathcal{L}_{\text{CF}}$ and $\mathcal{L}_{\text{ent}}$, and assuming predictive stability and behavioral degradation under intervention, the subset $C_{S_k}$ approximates a minimal sufficient set for predicting $Y$.*

**Theorem 2** (Lower Bound on the Behavioral Effect of Masking). *Let $\ell_f = \ell_{\text{pred}}(\hat{\mathbf{t}}_f, \mathbf{t}_{L+1:L+H})$ and $\ell_{cf} = \ell_{\text{pred}}(\hat{\mathbf{t}}_{cf}, \mathbf{t}_{L+1:L+H})$, and define the expected gap*

$$\Delta = \mathbb{E}[\ell_{cf} - \ell_f]. \tag{9}$$

*Then*

$$\mathcal{L}_{\text{CF}} \geq \log(1 + \exp(\Delta + \delta)), \qquad 0 < \frac{\partial \mathcal{L}_{\text{CF}}}{\partial(\ell_{cf} - \ell_f)} < 1. \tag{10}$$

**Overall Objective.** The final objective combines forecasting, counterfactual contrast, and sparsity:

$$\mathcal{L}_{\text{total}} = \mathbb{E}\left[\ell_{\text{pred}}(\hat{\mathbf{t}}_f, \mathbf{t}_{L+1:L+H})\right] + \lambda_{\text{CF}}\mathcal{L}_{\text{CF}} + \lambda_{\text{ent}}\mathcal{L}_{\text{ent}}, \tag{11}$$

with scalar weights $\lambda_{\text{CF}}, \lambda_{\text{ent}} \geq 0$.

## 4 EXPERIMENTS

**Selected Representative Datasets:** We evaluate our framework on a diverse set of benchmarks chosen for their relevance to context-aware forecasting. Instead of relying on arbitrary datasets, we focus on scenarios where auxiliary signals meaningfully influence future dynamics. The selected datasets cover various domains and exhibit different types of contextual dependencies, including environmental, behavioral, seasonal, and temporal factors. These include BJAQ, BitCoin, ILI, Electricity, Weather, ETT, Traffic, and Exchange. All experiments follow standard splits, look-back windows, and forecast horizons consistent with prior baselines. A full description and preprocessing details for all datasets are provided in Appendix C.

We compare our method against a comprehensive set of baselines across three major categories. (1) **Context-driven forecasting methods:** TGForecaster, ContextMatters, and TimXer, which explicitly model contextual influences. (2) **Foundation model-based approaches:** TabPFN, Chronos, MORAI, TEMPO, and GPT4TS, representing recent large-scale pretraining paradigms adapted to time series. (3) **Transformer-based models:** PatchTST, iTransformer, and DLinear, which serve as strong architectural baselines for sequence modeling. Full details of our experimental protocol, including training configurations and hyperparameter choices for all methods, are documented in Appendix C.

### 4.1 FULL-SHOT FORECASTING.

Our model achieves consistently strong performance across all datasets, as shown in Table 1. On benchmarks like BitCoin and ILI, where contextual features are stable and predictive, context-aware models such as ContextMatters and TimXer outperform backbone-only baselines like PatchTST and iTransformer, confirming the value of structured auxiliary signals. However, their advantages diminish on datasets like Traffic and Weather, where contextual inputs are noisy or weakly correlated. This reflects a key limitation: existing models rely on correlation without assessing context relevance, leading to overfitting on unstable signals. Large pre-trained models such as Chronos and TEMPO show better robustness under noise due to broad pretraining and scalable architectures. Yet they lack mechanisms to distinguish useful context from irrelevant signals, which limits adaptability across domains. In contrast, our model maintains high accuracy even when context quality varies.

Through query modulation and causal bias estimation, it learns to focus on structurally relevant signals while avoiding misleading patterns. These results suggest that accurate forecasting requires not just using context, but reasoning about which parts of it truly matter.

Table 1: Full-shot forecasting performance across nine datasets. We report MSE/MAE. The proposed method consistently outperforms strong baselines across diverse domains.

| Dataset | Ours | TGForecaster | ContextMatters | TimeXer | Chronos | TEMPO | GPT4TS | PatchTST | iTransformer |
|---|---|---|---|---|---|---|---|---|---|
| Bitcoin | **0.1519/0.2955** | 0.1618/0.3077 | 0.1612/0.3058 | 0.1636/0.3102 | 0.1648/0.3193 | 0.1701/0.3263 | 0.1773/0.3380 | 0.1784/0.3316 | 0.1798/0.3321 |
| ILI | **0.7205/0.6917** | 0.7585/0.6872 | 0.8874/0.7905 | 0.7501/0.7150 | 0.9107/0.7931 | 0.9972/0.8272 | 1.0857/0.8723 | 0.9175/0.8080 | 0.9138/0.7976 |
| Weather | **0.0016/0.0295** | 0.0017/0.0306 | 0.0017/0.0299 | 0.0017/0.0308 | 0.0017/0.0307 | 0.0017/0.0311 | 0.0017/0.0307 | 0.0017/0.0316 | 0.0017/0.0324 |
| Electricity | **0.3505/0.4161** | 0.4189/0.4677 | 0.3805/0.4352 | 0.3609/0.4209 | 0.3651/0.4484 | 0.3705/0.4361 | 0.3737/0.4374 | 0.3802/0.4353 | 0.3827/0.4429 |
| BJAQ | **1.3327/0.7829** | 1.6211/0.8407 | 1.5593/0.8474 | 1.3396/0.7867 | 1.3407/0.7902 | 1.3542/0.7921 | 1.3408/0.7922 | 1.4042/0.8040 | 1.4072/0.7973 |
| ETTh1 | **0.0749/0.2093** | 0.0811/0.2194 | 0.0817/0.2205 | 0.0781/0.2154 | 0.0769/0.2144 | 0.0773/0.2156 | 0.0775/0.2148 | 0.0805/0.2188 | 0.0807/0.2152 |
| ETTh2 | **0.1903/0.3439** | 0.1955/0.3487 | 0.2025/0.3540 | 0.1946/0.3467 | 0.1934/0.3440 | 0.1949/0.3461 | 0.1943/0.3452 | 0.1997/0.3513 | 0.1962/0.3495 |
| ETTm1 | **0.0528/0.1713** | 0.0558/0.1777 | 0.0561/0.1768 | 0.0540/0.1735 | 0.0539/0.1733 | 0.0531/0.1721 | 0.0537/0.1728 | 0.0562/0.1774 | 0.0544/0.1752 |
| ETTm2 | **0.1185/0.2546** | 0.1238/0.2626 | 0.1219/0.2573 | 0.1201/0.2570 | 0.1204/0.2572 | 0.1209/0.2571 | 0.1206/0.2573 | 0.1211/0.2566 | 0.1249/0.2637 |
| Traffic | **0.1948/0.2845** | 0.2765/0.3692 | 0.1984/0.2889 | 0.2034/0.2991 | 0.2228/0.3256 | 0.2245/0.3285 | 0.2327/0.3313 | 0.2444/0.3440 | 0.2484/0.3512 |
| Exchange | **0.4510/0.4613** | 0.4641/0.4675 | 0.4653/0.4708 | 0.4547/0.4657 | 0.4605/0.4716 | 0.4684/0.4718 | 0.4714/0.4739 | 0.4743/0.4799 | 0.4851/0.4852 |

## 4.2 ZERO-SHOT FORECASTING.

Table 2 reports the results under a leave-one-domain-out zero-shot setting. Foundation models like Chronos and TabPFN show strong transfer performance and outperform context-driven and Transformer-based baselines in most cases. Their advantage comes from large-scale pretraining, which helps capture domain-invariant temporal patterns. However, these models often overlook how context interacts with the target. Their focus on generic temporal dependencies limits their ability to adapt when the relevance or structure of context shifts across domains. For example, although Chronos and TabPFN remain stable overall, their performance degrades when unseen domains present new forms of context-target relationships. Our method maintains strong generalization while more effectively handling changes in context structure. By integrating instance-level causal filtering and context-aware attention modulation, the model identifies which contextual features remain useful under domain shift. As a result, it achieves consistent gains over both foundation and context-driven baselines. These results suggest that improving zero-shot forecasting requires more than scale or pretraining. It also demands mechanisms that can reason about context relevance under changing conditions. Refer to Appendix B for generalization analysis.

Table 2: Zero-shot forecasting performance under leave-one-domain-out evaluation. The proposed method shows strong generalization across unseen domains, outperforming diverse baselines.

| Model | Bitcoin | | ILI | | Weather | | Electricity | | BJAQ | |
|---|---|---|---|---|---|---|---|---|---|---|
| | MSE | MAE | MSE | MAE | MSE | MAE | MSE | MAE | MSE | MAE |
| **Ours** | **0.2243** | **0.3705** | **1.3241** | **0.9086** | **0.0014** | **0.0268** | **0.5215** | **0.5287** | **1.4709** | **0.8456** |
| TS Foundation Models: | | | | | | | | | | |
| TabPFN | 0.2285 | 0.3749 | 1.3279 | 0.9127 | 0.0015 | 0.0287 | 0.5323 | 0.5388 | 1.4924 | 0.8554 |
| Chronos | 0.2300 | 0.3754 | 1.4407 | 0.9425 | 0.0018 | 0.0310 | 0.5250 | 0.5316 | 1.4935 | 0.8558 |
| MORAI | 0.2314 | 0.3774 | 1.3435 | 0.9174 | 0.0016 | 0.0289 | 0.5274 | 0.5348 | 1.4978 | 0.8578 |
| TEMPO | 0.2331 | 0.3815 | 1.3346 | 0.9164 | 0.0014 | 0.0272 | 0.5299 | 0.5398 | 1.5044 | 0.8583 |
| GPT4TS | 0.2383 | 0.3897 | 1.3375 | 0.9145 | 0.0014 | 0.0273 | 0.5300 | 0.5388 | 1.5091 | 0.8602 |
| LLM Prompting: | | | | | | | | | | |
| GPT4o | 0.3278 | 0.4214 | 4.5177 | 1.5289 | 0.0681 | 0.1275 | 21.9941 | 3.8074 | 4.9123 | 1.7121 |
| GPT4o-mini | 0.2998 | 0.4427 | 2.7812 | 1.2151 | 0.0352 | 0.0850 | 17.0830 | 3.1637 | 4.7825 | 1.5830 |
| Llama3-8B | 1.0700 | 0.6971 | 6.5217 | 1.7829 | 0.6702 | 0.6040 | 20.5170 | 3.5183 | 2.1394 | 1.0047 |
| Mistral-7B | 0.9176 | 0.6575 | 6.1997 | 1.7083 | 0.9455 | 0.8280 | 28.7805 | 4.7294 | 5.0809 | 1.1821 |
| Context-driven Approach: | | | | | | | | | | |
| TGForecaster | 0.2540 | 0.4121 | 1.3766 | 0.9659 | 0.0016 | 0.0297 | 0.6407 | 0.6056 | 1.6457 | 0.9164 |
| ContextMatters | 0.2326 | 0.3830 | 1.4066 | 0.9207 | 0.0014 | 0.0273 | 0.5311 | 0.5405 | 1.5764 | 0.8807 |
| TimeXer | 0.2368 | 0.3880 | 1.4293 | 0.9280 | 0.0016 | 0.0296 | 0.5271 | 0.5378 | 1.5105 | 0.8611 |
| Transformer-based Approach: | | | | | | | | | | |
| PatchTST | 0.2340 | 0.3835 | 1.4094 | 0.9234 | 0.0014 | 0.0273 | 0.5267 | 0.5351 | 1.5791 | 0.8810 |
| iTransformer | 0.2362 | 0.3842 | 1.3786 | 0.9268 | 0.0014 | 0.0272 | 0.5316 | 0.5420 | 1.5330 | 0.8668 |

## 4.3 MODEL ANALYSIS

**Analysis of Learned Causal Structure** On the Bitcoin dataset, attention is inspected under three settings—(i) no structural bias, (ii) learned causal-bias weights, and (iii) attention after bias injection

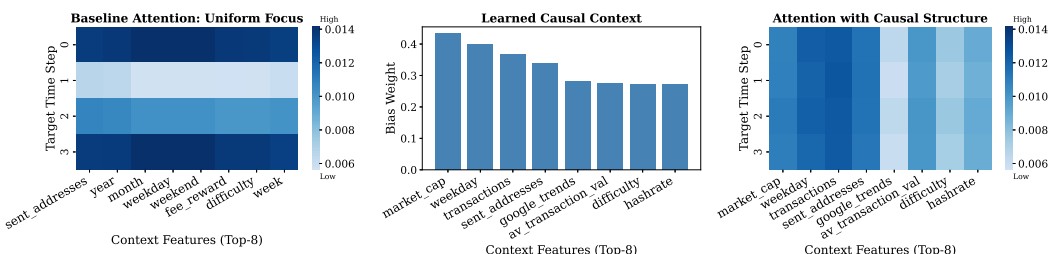

Figure 2: **Left**: Baseline attention shows uniform focus across context features. **Middle**: Learned causal bias highlights signals with strong predictive influence. **Right**: Attention becomes more selective after incorporating causal bias, emphasizing meaningful context.

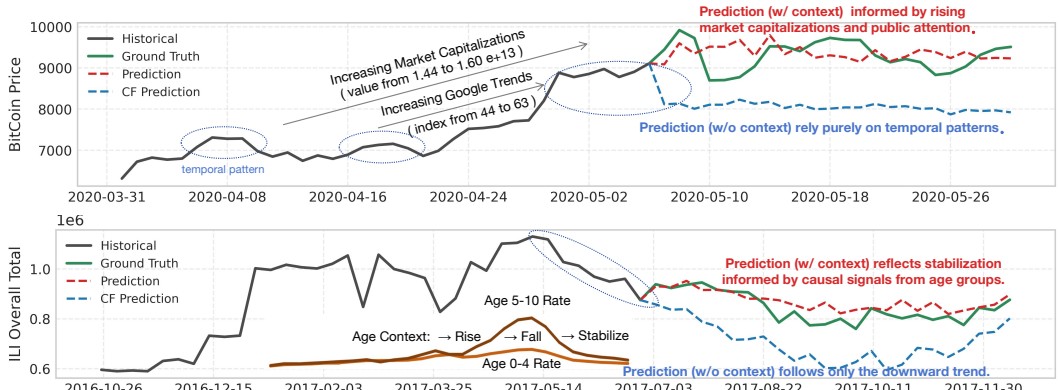

Figure 3: **Top**: In Bitcoin forecasting, the context-aware model captures the price surge using signals like market capacity, while the baseline fails to respond. **Bottom**: In ILI forecasting, age-specific infection rates help identify trend stabilization, whereas the baseline continues the declining trend.

(Figure 2). Without bias, attention is nearly uniform across the top-8 context features, frequently favoring temporal indicators (year/month/weekday) and exhibiting weak selectivity. The learned causal-bias map instead shows stable, time-consistent importance aligned with global causal salience (e.g., market capacity, Google Trends). After injecting this bias, attention shifts from superficial temporal cues toward causally relevant signals (e.g., market capacity, sent addresses), yielding sharper, semantically meaningful focus aligned with predictive drivers.

**Tracing Forecast to Causal Drivers**    To figure out how causal context contributes to prediction quality, we present two diagnostic cases that illustrate distinct modes of context influence (Figure 3). The first case involves Bitcoin price forecasting during a period of sharp market shift in May 2020. At this point, contextual signals such as rising market capitalization (from 1.44 to $1.60 \times 10^{13}$) and increasing Google Trends index (from 44 to 63) reflected heightened public attention and speculative activity. The context-aware model captures the upward trend by incorporating macro-level indicators as causal drivers in its prediction. In contrast, the model without context follows only temporal trends and misses the structural change, resulting in a flat forecast. This highlights the importance of causal context in providing early signals that precede and help explain shifts in the prediction target. A second case on influenza forecasting shows that internal context, such as age-specific infection rates, can reveal early causal signals preceding trend shifts. Full analysis is provided in Appendix C.

**Adaptive Context Utilization and the Role of Counterfactual Intervention**    On query modulation, Figure 4 (top) reports the distribution of learned modulation scalars across five datasets. These scalars rescale the query representation before cross-attention, thereby adjusting context integration strength. Patterns are dataset-specific: Electricity peaks near 0.45, indicating increased reliance on temporal cues (e.g., hour, weekday); Weather concentrates near zero or negative, suppressing ex-

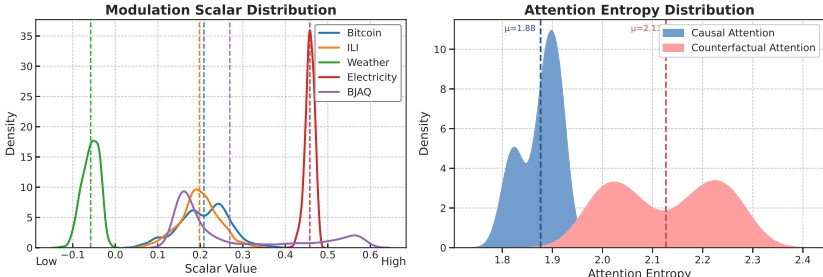

Figure 4: **Left**: Query modulation scalar distributions across five datasets. The model adjusts the strength of context based on the usefulness of input signals. **Right**: Attention entropy with and without causal bias. Causal guidance leads to lower entropy and more focused attention.

ternal inputs when target autocorrelation is sufficient; Bitcoin, ILI, and BJAQ exhibit broader or multimodal shapes, consistent with selective, data-dependent usage. Overall, the model regulates context according to structural relevance.

For counterfactual intervention, attention entropy is compared with and without learned causal bias (Figure 4, bottom). Causal guidance yields a narrower distribution centered at lower entropy, whereas removing the bias markedly increases entropy ($p < 1\mathrm{e}^{-5}$). These results indicate sharper, more selective attention under causal bias, with the counterfactual loss further discouraging focus on spurious signals.

### 4.4 ABLATION STUDY

Ablation results (Table 3) show that **w/o Context** yields the largest overall degradation, indicating temporal modeling alone is insufficient and validating the centrality of context in real-world settings. The **r/p Vanilla Attention** baseline—plain cross-attention without reasoning—consistently underperforms the full model, underscoring that effective integration requires causal assessment rather than mere access to signals. Among structural components, removing **w/o Causal Bias** causes the largest drop, highlighting the need to assign stable, global importance to context dimensions for robust generalization under noisy or shifting conditions. Eliminating **w/o Query Modulation** also degrades accuracy, notably on Bitcoin and ILI, confirming that aligning temporal queries with global context semantics improves attention selectivity and reduces redundancy. Finally, **w/o Counterfactual Loss** produces a moderate decline—especially under unstable context–target relations—indicating that counterfactual supervision sharpens focus by discouraging attention to spurious features and improving consistency between attribution and predictive behavior. Collectively, these results support the design choice of causal-aware modules for adaptive, reliable context integration.

## 5 CONCLUSION

In this work, we introduce a causally grounded framework for context-aware time series forecasting. Contextual signals are integrated through a cross-attention mechanism guided by semantic query modulation and counterfactual interventions. Extensive experiments on real-world benchmarks demonstrate that our model delivers superior forecasting accuracy and provides better interpretability. Moving forward,

Table 3: Ablation Study: MAE metric across five datasets after removing each component.

| Model | BTC | ILI | WTH | ELE | BJAQ |
|---|---|---|---|---|---|
| **Ours** | **0.2955** | **0.6928** | **0.0295** | **0.4161** | **0.7835** |
| w/o CF | 0.2974 | 0.6993 | 0.0301 | 0.4176 | 0.7881 |
| w/o QM | 0.2987 | 0.6984 | 0.0298 | 0.4173 | 0.7850 |
| w/o CB | 0.3006 | 0.6933 | 0.0297 | 0.4174 | 0.7855 |
| w/o CT | 0.3368 | 0.8661 | 0.0312 | 0.4358 | 0.7905 |
| r/p VA | 0.3076 | 0.7261 | 0.0305 | 0.4258 | 0.7893 |

we plan to explore several directions. One promising avenue is to leverage LLMs for context interpretation and factor abstraction. In addition, retrieval-augmented generation (RAG) may offer a principled way to incorporate external knowledge on demand.

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

APPENDIX

This appendix complements the main paper with formal theoretical foundations (Section A), an in-depth analysis of generalization behavior and limitations (Section B), comprehensive experimental setup and extended benchmark results (Section C), and a curated overview of related literature (Section D).

## A  THEORETICAL ANALYSIS AND PROOFS

### A.1  INFORMATION BOTTLENECK DERIVATION FOR CONTEXT-AWARE QUERY MODULATION

We provide a formal derivation for the information-theoretic motivation behind Context-Aware Query Modulation, based on the Information Bottleneck (IB) principle.

Let $C$ denote the causal context, $Y$ the prediction target, and $T$ the intermediate modulated query representation. The goal is to obtain a compressed representation $T$ that retains maximal information about $Y$ while discarding irrelevant aspects of $C$.

Formally, the IB objective seeks to optimize:

$$\min_{p(T|C)} \; I(T;C) - \beta I(T;Y),$$

where $I(\cdot;\cdot)$ denotes mutual information and $\beta > 0$ is a trade-off parameter.

We expand the mutual information terms:

$$I(T;C) = \mathbb{E}_{p(C,T)}\left[\log\frac{p(T|C)}{p(T)}\right], \quad I(T;Y) = \mathbb{E}_{p(T,Y)}\left[\log\frac{p(Y|T)}{p(Y)}\right].$$

Thus, the IB objective becomes:

$$\mathcal{L}_{\text{IB}} = \mathbb{E}_{p(C)}\left[\mathbb{E}_{p(T|C)}\left[\log\frac{p(T|C)}{p(T)}\right]\right] - \beta\mathbb{E}_{p(T,Y)}\left[\log\frac{p(Y|T)}{p(Y)}\right].$$

In practice, optimizing $p(T|C)$ exactly is intractable. Instead, we parameterize the mapping from $C$ to $T$ via a deterministic function $T = h(C)$, corresponding to the modulation operation:

$$T = h(C) = \mathbf{Q}_{\text{mid}} \odot (1 + f(C)).$$

Under deterministic mapping, $p(T|C)$ degenerates to a Dirac delta, and thus $I(T;C) = H(T) - H(T|C) = H(T)$, meaning we can interpret minimizing $I(T;C)$ as minimizing the entropy of $T$. That is, we aim to make $T$ as compressed (low-entropy) as possible while maintaining task performance.

Meanwhile, the second term $I(T;Y)$ encourages $T$ to retain predictive information about $Y$.

Therefore, Context-Aware Query Modulation operationalizes an implicit minimization of the following practical surrogate:

$$\mathcal{L}_{\text{practical}} = H(T) - \beta I(T;Y),$$

where $H(T)$ can be indirectly controlled by the degree of modulation, and $I(T;Y)$ is maximized via standard task loss optimization.

Thus, query modulation introduces a selective information bottleneck at the query formation stage, balancing compression and predictive power.

### A.2  THEORETICAL JUSTIFICATION FOR APPROXIMATE MINIMAL SUFFICIENCY

In this section, we provide a theoretical justification for why our causal structure bias $\mathbf{B}_{\text{causal}}$—learned through counterfactual supervision and entropy regularization—converges to an approximate *minimal sufficient subset* of context variables for predicting the target sequence.

**Background: Minimal Sufficient Subset**

Let $C = \{C_1, C_2, \ldots, C_N\}$ denote the full set of context variables, and let $Y$ denote the target variable. A subset $S \subseteq \{1, \ldots, N\}$ is called *sufficient* if:

$$P(Y \mid C_S) = P(Y \mid C),$$

and *minimal* if no proper subset of $S$ is also sufficient.

In general, discovering minimal sufficient subsets is intractable without knowledge of the true data-generating process. However, under structural learning constraints and appropriate inductive biases, it is possible to approximate such subsets via differentiable mechanisms.

**Setup: Causal Structure Bias and Top-$k$ Selection**

Our model learns a real-valued bias vector $\mathbf{B}_{\text{causal}} \in \mathbb{R}^N$ through a structural equation-style module:

$$\mathbf{B}_{\text{causal}} = \psi_{\text{SEM}}(\mathbf{Z}_c),$$

where $\mathbf{Z}_c \in \mathbb{R}^{T' \times D}$ is the encoded context representation. At inference or training time, the model selects the top-$k$ indices based on $\mathbf{B}_{\text{causal}}$:

$$S_k = \text{TopK}(\mathbf{B}_{\text{causal}}) \subseteq \{1, \ldots, N\},$$

and masks all other entries during cross-attention.

**Training Objectives and Behavioral Alignment**

We optimize three terms jointly:

- **Prediction loss** $\mathcal{L}_{\text{pred}}$: standard forecasting loss (e.g., MSE).
- **Counterfactual loss** $\mathcal{L}_{\text{CF}}$: measures behavioral divergence under removal of top-$k$ variables:
$$\mathcal{L}_{\text{CF}} = \log\left(1 + \exp\left(\mathcal{L}_{\text{pred}}(\hat{y}_{\text{cf}}) - \mathcal{L}_{\text{pred}}(\hat{y}_{\text{f}}) + \delta\right)\right),$$
where $\hat{y}_{\text{f}}$ is the factual prediction, $\hat{y}_{\text{cf}}$ is the prediction under masking of $S_k$.
- **Entropy loss** $\mathcal{L}_{\text{ent}}$: encourages sparse structural bias:

$$\mathcal{L}_{\text{ent}} = -\sum_{i=1}^{N} \alpha_i \log \alpha_i, \quad \text{where } \alpha_i = \text{Softmax}(\mathbf{B}_{\text{causal}})_i.$$

**Main Result**

We now present the main theoretical justification.

**Theorem 3** (Approximate Minimal Sufficiency under Structural Regularization)**.** *Let $S_k = TopK(\mathbf{B}_{causal})$ be the selected context support set. Suppose the following hold:*

*(i)* ***Prediction Equivalence:*** *$\mathbb{E}[\mathcal{L}_{pred}(C_{S_k})] \approx \mathbb{E}[\mathcal{L}_{pred}(C)]$;*

*(ii)* ***Counterfactual Divergence:*** *$\mathbb{E}[\mathcal{L}_{pred}(\hat{y}_{cf}) - \mathcal{L}_{pred}(\hat{y}_f)] > \delta > 0$;*

*(iii)* ***Entropy Minimization:*** *$\mathbb{H}(Softmax(\mathbf{B}_{causal})) \to \min$.*

*Then $C_{S_k}$ is an approximate minimal sufficient subset of $C$ for predicting $Y$.*

**Proof Sketch**

- Assumption (i) ensures that $C_{S_k}$ is **sufficient**: it retains all predictive information.
- Assumption (ii) ensures **functional salience**: removing $S_k$ leads to performance drop, ruling out redundancy.
- Assumption (iii) ensures **minimality**: under entropy minimization, $S_k$ becomes compact—i.e., no larger subset than necessary.

Hence, together the losses $\mathcal{L}_{\text{CF}}$ and $\mathcal{L}_{\text{ent}}$ not only regularize structure, but also construct a predictive path grounded in causal attribution. The model is incentivized to select the smallest subset of context variables that are both sufficient for accurate forecasting and impactful under counterfactual intervention—thereby approximating minimal sufficient structure in an end-to-end differentiable manner.

### A.3 PROOF OF THEOREM 1 (LOWER BOUND ON CAUSAL EFFECT OF STRUCTURAL BIAS)

**Theorem 1 (restated).** Let $\Delta = \mathbb{E}[\ell_{\mathrm{cf}} - \ell_{\mathrm{f}}]$ denote the expected increase in prediction loss under counterfactual intervention (i.e., masking high-scoring features in $B_{\mathrm{causal}}$). Then the counterfactual divergence loss $\mathcal{L}_{\mathrm{CF}}$ satisfies:

$$\mathcal{L}_{\mathrm{CF}} \geq \log(1 + \exp(\Delta + \delta)), \quad \text{with } 0 < \frac{\partial \mathcal{L}_{\mathrm{CF}}}{\partial(\ell_{\mathrm{cf}} - \ell_{\mathrm{f}})} < 1$$

**Proof.** Recall that the counterfactual loss is defined as:

$$\mathcal{L}_{\mathrm{CF}} = \log(1 + \exp(\ell_{\mathrm{cf}} - \ell_{\mathrm{f}} + \delta))$$

Let us denote $z = \ell_{\mathrm{cf}} - \ell_{\mathrm{f}} + \delta$. Then $\mathcal{L}_{\mathrm{CF}} = \log(1 + \exp(z))$ is the softplus function, which is convex and monotonically increasing in $z$.

To understand the expected behavior of this loss, we take the expectation over $z$:

$$\mathbb{E}[\mathcal{L}_{\mathrm{CF}}] = \mathbb{E}[\log(1 + \exp(z))] = \mathbb{E}[\log(1 + \exp(\ell_{\mathrm{cf}} - \ell_{\mathrm{f}} + \delta))]$$

By Jensen's inequality, applied to the convex softplus function, we obtain:

$$\mathbb{E}[\mathcal{L}_{\mathrm{CF}}] \geq \log(1 + \exp(\mathbb{E}[\ell_{\mathrm{cf}} - \ell_{\mathrm{f}}] + \delta)) = \log(1 + \exp(\Delta + \delta))$$

This inequality provides a lower bound on the expected counterfactual loss in terms of the average causal performance gap $\Delta$. Intuitively, it formalizes the idea that if masking causally important features causes prediction performance to drop, then $\mathcal{L}_{\mathrm{CF}}$ must be correspondingly large.

For the second part of the theorem, we examine the sensitivity of the loss with respect to the causal gap:

$$\frac{\partial \mathcal{L}_{\mathrm{CF}}}{\partial(\ell_{\mathrm{cf}} - \ell_{\mathrm{f}})} = \frac{e^z}{1 + e^z} = \sigma(z)$$

where $\sigma(\cdot)$ is the sigmoid function.

Since the sigmoid function satisfies $\sigma(z) \in (0, 1)$ for all $z \in \mathbb{R}$, we have:

$$0 < \frac{\partial \mathcal{L}_{\mathrm{CF}}}{\partial(\ell_{\mathrm{cf}} - \ell_{\mathrm{f}})} < 1$$

This shows that $\mathcal{L}_{\mathrm{CF}}$ responds smoothly to changes in the causal performance gap, avoiding gradient explosion or vanishing. This property is important for optimization stability.

∎

## B METHOD

### B.1 GENERALIZATION ANALYSIS

Generalizing to unseen domains with limited or partial contextual information remains a fundamental challenge in context-aware forecasting. In our setting, each test dataset only exposes a subset of contextual variables due to masking. Despite this constraint, our model demonstrates strong cross-domain performance. This section analyzes the core factors behind this generalization behavior, focusing on how the model interacts with input context and learns transferable structural patterns during training.

**Selective reasoning within retained context.** The masking mechanism filters out unsupported context channels per dataset, but the retained channels still vary in their causal influence. Our model does not assume uniform utility across context inputs. Instead, it learns to assess which of the available signals contribute meaningfully to the prediction. This instance-level selection is achieved through semantic query modulation and causal bias estimation, which adjust attention weights based on the input content. As a result, even when only a small context subset is visible, the model can suppress weak signals and focus on those that are predictive.

**Input-adaptive design.** Our attention modulation and causal reasoning modules operate directly on the input embeddings of available context. These mechanisms do not rely on fixed context identities, but instead compute relevance based on the actual representations. This allows the model to adapt dynamically to any valid context subset, making it robust to missing channels and compatible with unseen combinations during inference.

**Cross-domain structural priors.** During training, the model is exposed to diverse context-target relationships across multiple datasets. This cross-domain supervision enables it to internalize structural patterns that characterize when and how different types of context tend to influence predictions. At test time, the model leverages these priors to reason over newly observed context structures, even when the specific dataset has never been seen before. The causal bias module generalizes particularly well, as it is trained to infer relevance without relying on dataset-specific assumptions.

**Conclusion.** The model's generalization capability comes from its ability to reason within the visible context subset, adapt its inference based on input semantics, and reuse structural knowledge learned from prior datasets. These properties make it robust to context variability and domain shift, even under partial observability.

## B.2 LIMITATION ANALYSIS

While our proposed framework demonstrates strong generalization and interpretability across diverse datasets, certain aspects of the design suggest potential directions for further refinement. First, the causal inductive bias is derived from learned structural equations, which, while effective in identifying globally influential context features, may be sensitive to the quality and stability of contextual inputs across domains. In scenarios where causal relationships are weak or highly dynamic, additional mechanisms may be needed to ensure robust adaptation.

Second, although counterfactual interventions provide behavioral supervision and improve attribution reliability, they are currently applied during training only. Extending such reasoning mechanisms to inference-time adaptation or online settings may further enhance robustness under distribution shifts.

Third, while we adopt a unified encoder for heterogeneous context formats, the representation space remains implicitly learned. Exploring more explicit cross-modal alignment or incorporating external knowledge (e.g., via retrieval-augmented generation) may improve performance in settings with sparse or weak contextual signals.

Finally, our current experiments focus on structured forecasting benchmarks with available context annotations. Broader validation on real-world deployments with irregular or unstructured context remains an exciting direction for future exploration.

## C EXPERIMENT

### C.1 DATASETS DESCRIPTION

We evaluate the proposed framework on a comprehensive suite of publicly available benchmark datasets, selected to capture a broad range of contextual factors relevant to real-world time series forecasting. These datasets cover diverse application areas such as air quality, finance, epidemiology, energy, and meteorology, and are characterized by the presence of rich auxiliary context that significantly influence future dynamics. A detailed description of each dataset used in our experiments is provided below.

**BJAQ (Beijing Air Quality)** The BJAQ dataset contains hourly records of air pollutants and meteorological conditions collected from 12 monitoring sites across Beijing, sourced from the Beijing Municipal Environmental Monitoring Center and the China Meteorological Administration. The dataset spans from March 2013 to February 2017 and includes six air quality indicators and six meteorological variables (e.g., temperature, wind direction, humidity). Missing numeric values are imputed by mean replacement, and categorical variables such as wind direction are filled with an `unknown` token. After applying a sliding window (length 96, stride 24), we obtain 12,166 training

samples, 1,537 validation samples, and 1,525 test samples. This dataset exemplifies environmental context dependence, where features like wind speed and temperature modulate air quality evolution.

**BitCoin (BTC)** The BTC dataset, from the Monash Time Series Forecasting Repository, consists of daily closing prices for Bitcoin from 2010 to 2021, augmented with 18 exogenous features that influence price movements. These include blockchain metrics (e.g., hash rate, block size), mining difficulty, and behavioral indicators such as Google search trends and social interest around "Bitcoin." Despite partial missingness in context variables, the dataset offers rich behavioral context for financial forecasting. We use a 7:1:2 train/val/test split with univariate targets.

**ILI (Influenza-Like Illness)** Collected by the CDC between 2002 and 2021, the ILI dataset reports weekly flu case counts and incidence rates per 100,000 individuals across different U.S. regions. The dataset provides four types of contextual features: calendar indicators (e.g., week of year), age-specific case rates, regional tags, and flu strain types. Its long temporal span and fine granularity make it ideal for evaluating seasonal and epidemiologically sensitive models. This dataset is commonly used for public health forecasting and policy modeling.

**Weather** This dataset contains 21 meteorological variables spanning multiple temporal granularities. With over 25,000 initial records sampled every 10 minutes, it is resampled into hourly, daily, and weekly intervals to support multi-resolution forecasting. Features include temperature ($T$), pressure ($P$), vapor metrics (e.g., $VP_{max}$, $VP_{def}$), humidity, and solar radiation. Key features like $T$, $P$, and $T_{dew}$ retain high-resolution sampling, while others are aggregated to reduce noise. Its multiscale nature makes it suitable for causal analysis across time granularities.

**Electricity** This dataset comprises hourly electricity consumption data from 370 users, recorded from 2011 to 2015. Each time series captures monthly, weekly, and daily demand fluctuations, with contextual features including calendar variables such as day of week and holidays. The dataset has 540,200 records in total, with standard splits adopted for forecasting. Its regular temporal structure and demand-cycle patterns offer an ideal testbed for learning calendar-sensitive causal signals.

Additional datasets such as ETT (Electricity Transformer Temperature), Traffic and Exchange are also used in our experiments and follow standard preprocessing pipelines. These datasets exhibit contextual dependencies ranging from load-driven temperature effects and temporal autocorrelations to economic and promotional events.

All datasets are standardized with respect to input sequence length, forecasting horizon, and normalization settings to ensure comparability.

## C.2 CONTEXTUAL FEATURE SPECIFICATION

To fully harness the potential of context-aware forecasting, we systematically incorporate auxiliary variables that provide explanatory signals beyond the target sequence. We categorize all contextual inputs into two major types:

- **Dataset-Specific Context**: Variables that are unique to each dataset and reflect domain-specific conditions, such as financial indicators in Bitcoin or pollutant concentrations in air quality prediction.
- **Shared Temporal Context**: A unified set of time-derived features applicable across all datasets, capturing periodic patterns, calendar structures, holidays, and sociocultural events.

This two-tiered organization enables our model to balance localized, task-specific information with global, transferable temporal priors. The following sections detail the contextual variables used in our framework, organized by their origin and scope.

### C.2.1 CONTEXT VARIABLE SPECIFICATION

Our framework is designed to leverage rich contextual signals beyond the target variable itself. To ensure model generalizability and meaningful context reasoning, we carefully identify and utilize

auxiliary variables available in each dataset. In this section, we detail the contextual features adopted for each benchmark dataset. We exclude the forecasting target (denoted as `OT`) and focus solely on variables that serve as inputs for context-aware modeling.

**Bitcoin**  The Bitcoin dataset includes a variety of financial and behavioral indicators that influence price dynamics. The contextual variables comprise transactional activity metrics (e.g., `sent_addresses`, `transactions`, `median_transaction_size`), blockchain-level statistics (e.g., `hashrate`, `block_size`, `difficulty`), market indicators (e.g., `market_cap`, `mining_profitability`, `fee_reward`), and external behavioral signals such as `google_trends` and `active_addresses`. The `date` feature is used to derive temporal patterns but excluded as a raw input.

**ILI (Influenza-Like Illness)**  The ILI dataset provides epidemiological context across regions and time. Contextual variables include weighted and unweighted influenza-like illness (ILI) rates (`% WEIGHTED ILI`, `%UNWEIGHTED ILI`), age-group-specific incidence rates (e.g., `AGE 0-4`, `AGE 5-24`), aggregate statistics such as `ILITOTAL`, and healthcare infrastructure indicators like `NUM. OF PROVIDERS`.

**Weather**  The weather dataset contains high-resolution meteorological observations across multiple time granularities. Context variables include atmospheric pressure (`p`), air and dew temperatures (`T`, `Tpot`, `Tdew`), relative humidity (`rh`), various vapor pressure metrics (`VPmax`, `VPact`, `VPdef`), and other environmental measures like `rain`, `SWDR` (shortwave downward radiation), `PAR`, `Tlog`, and wind conditions (`wv`, `wd`). These variables offer rich contextual cues for modeling dynamic atmospheric changes.

**Electricity**  This dataset consists of electricity consumption data from 370 users, each represented by a distinct time series. The contextual inputs comprise 319 anonymized variables that encode individual consumption patterns, calendar effects, and regional behaviors. These context features capture both user-level temporal profiles and broader demand trends, making them ideal for calendar-sensitive modeling.

**BJAQ (Beijing Air Quality)**  The BJAQ dataset provides meteorological and pollutant-related context signals that shape air quality fluctuations. Contextual features include six air pollutant concentrations (`PM10`, `SO2`, `NO2`, `CO`, `O3`), temperature-related indicators (`TEMP`, `PRES`, `DEWP`), rainfall amount (`RAIN`), and wind direction/speed variables spanning 16 compass points and `WSPM` (wind speed in m/s).

**ETT (Electricity Transformer Temperature)**  The ETT dataset contains contextual features representing electricity transformer operational conditions. These include high-, medium-, and low-usage features for both load and frequency: `HUFL`, `HULL`, `MUFL`, `MULL`, `LUFL`, and `LULL`. These provide operational context regarding thermal conditions that affect temperature evolution over time.

**Traffic**  The Traffic dataset contains 860 contextual variables corresponding to hourly traffic occupancy rates from different lanes and sensors. Each series reflects dynamic local conditions such as traffic congestion, time-of-day effects, and regional traffic flow patterns. These are used as contextual input for short-term traffic forecasting.

**Exchange Rate**  The Exchange dataset consists of six contextual variables representing daily exchange rates between various currencies. These include normalized exchange rates for currency pairs indexed as `0` through `5`, reflecting global market dynamics influenced by geopolitical and macroeconomic factors. The variable `date` is excluded as a standalone feature but may be indirectly encoded via temporal position embeddings.

C.2.2  SHARED TEMPORAL CONTEXT

In addition to dataset-specific contextual features, we design a shared temporal context module applicable across all datasets. This module extracts a rich set of time-aware features that encode periodicity, calendar structure, and sociocultural signals. These features are automatically derived

from the `date` column of each dataset and form a general-purpose, domain-agnostic temporal prior for forecasting. The extracted temporal context can be grouped into three categories:

**Cyclic Time Encodings**    We apply sine/cosine transformations to model natural periodic patterns at various granularities:

- **Month of Year**: `month_sin`, `month_cos` ($\frac{2\pi m}{12}$)
- **Week of Month**: `week_of_month_sin`, `week_of_month_cos`
- **Weekday of Week**: `weekday_sin`, `weekday_cos` ($\frac{2\pi d}{7}$)
- **Hour of Day**: `hour_sin`, `hour_cos` (if hourly data)
- **Minute of Hour**: `minute_sin`, `minute_cos` (if sub-hourly data)

**Calendar and Seasonal Indicators**    To support long-term temporal structures, we introduce:

- **Normalized Year**: `year_scaled`, min-max normalized within dataset span
- **Season Multi-Hot**:    `season_spring`, `season_summer`, `season_autumn`, `season_winter`
- **Weekend Indicator**: `is_weekend`
- **Quarter Indicator**: `is_quarter_start`, `is_quarter_end`
- **Academic Calendar**: `spring_semester`, `summer_break`, `fall_semester`

**Holiday and Social Event Signals**    To encode sociocultural context that may affect target dynamics (e.g., retail spikes, reduced mobility), we include:

- **Holiday Indicator**: `is_holiday`, based on national calendars (e.g., US, CN)
- **Social Event Indicator**: `is_social_event`, covering both Gregorian (e.g., `Black Friday`, `Christmas`) and lunar holidays (e.g., `Spring Festival`, `Qixi Festival`)
- **Multi-hot Social Event Types**:    e.g.,    `social_event_Double_11`, `social_event_Lantern_Festival`, etc.

**Implementation and Usage**    These temporal features are automatically generated via a standardized module (`ContextTimeFeatures`), implemented in Python. This module supports flexible customization and is fully dataset-agnostic. The generated features are concatenated with other context vectors and jointly fed into the model's context encoder. This rich and unified temporal encoding enables the model to better capture seasonality, detect event-driven anomalies, and generalize across domains with varying time structures.

## C.3    Handling and Definition of Context

**1. Supporting Literature for a Unified Notion of Context.**    The term *context* is consistent with recent research in time–series forecasting that systematically models external information—ranging from covariates to textual signals—within a unified framework. A series of studies explicitly unifies all *non-series auxiliary signals* under the umbrella term *context*. Within this convention, *context* encompasses categorical or continuous covariates, time-varying exogenous variables, and free-form text; treating these heterogeneous sources jointly has become standard practice in contemporary forecasting research.

**2. Processing of Heterogeneous Context Types.**    All context types are handled within a unified representation framework. Numerical variables are standardized and linearly projected into continuous embeddings. Categorical variables are transformed by embedding lookups derived from multi-hot encodings. Textual variables are encoded by a lightweight pre-trained language model, followed by pooling to obtain fixed-dimensional vectors. The resulting representations are projected into a shared embedding space to support seamless integration via attention-based modules.

Table 4: Supporting literature for unified context and how each paper defines/uses context.

| Evidence | How the paper defines or uses context |
|---|---|
| **Context Is Key (CiK) benchmark** Williams et al. (2024) — Explicitly introduces *context-aided forecasting*, and unifies "any information related to forecasting and complementary to historical observations" under the notation $C$. | Establishes the generic formulation $P(X_F \mid X_H, C)$, explicitly allowing any side information to serve as *context*. |
| **ContextFormer** Chattopadhyay et al. (2024) — Refers to "categorical, continuous, time-varying, and even textual information" as *rich multimodal context*, emphasizing holistic integration into forecasting models. | Shows that disparate covariates and text are treated as a single contextual modality. |
| **ContexTST** Hong et al. (2025) — Frames "contextual information" as *domain anchors* that guide cross-domain generalization, automatically generating textual descriptions to provide "complementary knowledge" beyond raw series values. | Reinforces the notion that external textual (and other) signals are unified as context for transfer learning. |

Together, these works justify the terminology: referring to *any auxiliary signal that is not the target time series itself* as **context** is now an accepted convention.

- **Numerical context:** examples include mobility rates and transaction volumes; each feature is first normalized and then mapped into the context embedding matrix $\mathbf{C}$ through a dedicated linear projection layer.

- **Categorical context:** examples include region identifiers and age groups; categories are converted into embeddings using multi-hot encoding, with embeddings shared across time steps.

- **Textual context:** examples include policy documents or news articles; a lightweight encoder (e.g., MiniLM) tokenizes and embeds the text, and mean pooling over token embeddings yields a fixed-dimensional vector at either the sample or time-step level, depending on availability.

- **Shared temporal context:** as detailed in Appendix C.2.2, time-related categorical features (e.g., weekday, month) are represented via multi-hot encodings, while continuous time features (e.g., minute of day) are represented via sinusoidal positional encodings.

**3. Categorization of Context Variables Across Datasets.** A catalog of dataset-specific *context* variables is provided below, grouped by *type* (covariates vs. domain-specific metadata) together with concise descriptions and justifications.

### C.3.1 BJAQ (Beijing Air Quality)

*Scope.* Hourly air pollutant concentrations and meteorological data from 12 Beijing stations (2013–2017). Pollution is strongly affected by wind speed and temperature; a classical environmental forecasting task. Context variables are *internal*.

**Covariates** *Variables:* TEMP, DEWP, PRES, WSPM, RAIN
    *Description:* Measurable meteorological variables directly used for pollution prediction
    *Justification:* Dynamic, continuous measurements; typical numeric covariates
**Domain-specific metadata** *Variables:* PM10, SO2, NO2, CO, O3, Wind Direction

*Description:* Pollution concentrations and wind direction reflecting structural environmental information

*Justification:* Though time-varying, they express domain-specific pollution status

### C.3.2 BITCOIN

*Scope.* Daily Bitcoin closing prices (2010–2021) with blockchain, mining, and social signals; representative financial forecasting benchmark. Context variables are *internal*.

**Covariates** *Variables:* `Transactions, Block size, Hashrate, Difficulty`
  *Description:* On-chain behavior and technical indicators influencing price
  *Justification:* Used directly as time-varying numerical covariates
**Domain-specific metadata** *Variables:* `Google Trends, Active Addresses, Mining Profitability`
  *Description:* Social interest and domain-level semantic indicators
  *Justification:* Reflect long-term structural or social drivers beyond pure numeric trends

### C.3.3 ILI (INFLUENZA-LIKE ILLNESS)

*Scope.* Weekly influenza case counts and epidemiological metadata across US regions (age-specific rates, strain types, regional tags). Context variables are *internal*.

**Covariates** *Variables:* `AGE 0--4, AGE 5--24, ..., AGE 65+`
  *Description:* Age-stratified infection rates indicating spread dynamics
  *Justification:* Time-varying signals used directly for forecasting
**Domain-specific metadata** *Variables:* `REGION, STRAIN TYPE, NUM. OF PROVIDERS`
  *Description:* Spatial tags, virus strain labels, and healthcare resource metadata
  *Justification:* Static or categorical structure, not dynamic covariates

### C.3.4 WEATHER

*Scope.* High-frequency meteorological measurements (every 10 minutes) resampled to hourly/daily/weekly series for multiscale forecasting. Context variables are *internal*.

**Covariates** *Variables:* `T, RH, P, VPmax, VPdef, SWDR, etc.`
  *Description:* Environmental variables capturing atmospheric dynamics
  *Justification:* All variables are continuous and time-varying
**Domain-specific metadata** *Variables:* `None`
  *Description:* —
  *Justification:* No static or structural metadata present

### C.3.5 ETT (ELECTRICITY TRANSFORMER TEMPERATURE)

*Scope.* Transformer thermal dynamics under different load/frequency conditions, including high-/mid/low usage indicators; common in power systems modeling. Context variables are *internal*.

**Covariates** *Variables:* `HUFL, HULL, MUFL, MULL, LUFL, LULL`
  *Description:* Transformer load and frequency across usage levels
  *Justification:* Time-varying measurements directly linked to target dynamics
**Domain-specific metadata** *Variables:* `None`
  *Description:* —
  *Justification:* No structural or categorical context is provided

### C.3.6 TRAFFIC

*Scope.* Hourly occupancy rates from 860 urban road sensors, reflecting traffic flows and congestion. Context variables are *internal*.

**Covariates** *Variables:* 860-dimensional occupancy sequences
    *Description:* High-dimensional dynamic sensor inputs
    *Justification:* Each sensor represents a separate time-varying input channel
**Domain-specific metadata** *Variables:* Sensor ID (implicit)
    *Description:* Reflects spatial layout of roads/regions
    *Justification:* Not directly used in modeling, but represents latent structure

### C.3.7 EXCHANGE RATE

*Scope.* Daily exchange rates for multiple currency pairs used to model global financial trends. Context variables are *internal*.

**Covariates** *Variables:* Exchange rate sequences (pairs 0–5)
    *Description:* Daily variations between currencies
    *Justification:* Standard numerical time series
**Domain-specific metadata** *Variables:* None
    *Description:* —
    *Justification:* No geopolitical or structural metadata provided

### C.3.8 SHARED TEMPORAL CONTEXT (ALL DATASETS)

*Scope.* A shared temporal context module across datasets capturing seasonality and sociocultural factors (periodic encodings, holiday indicators, academic calendars). Context variables are *internal*.

**Covariates** *Variables:* `hour_sin`, `month_cos`, `is_weekend`, `is_holiday`, etc.
    *Description:* Time-based periodic and event features
    *Justification:* Derived from timestamps; universally applicable
**Domain-specific metadata** *Variables:* None
    *Description:* —
    *Justification:* No external or structural metadata involved

## C.4 EVALUATION METRICS

To evaluate forecasting performance, we adopt two widely used and interpretable metrics: Mean Squared Error (MSE) and Mean Absolute Error (MAE). These metrics quantify the discrepancy between predicted sequences and ground-truth observations, providing complementary views of accuracy and robustness.

- **Mean Squared Error (MSE)** MSE computes the average of squared differences between the predicted values $\hat{X}_h$ and the ground-truth values $X_h$ over the prediction horizon $H$:

$$\text{MSE} = \frac{1}{H} \sum_{h=1}^{H} (X_h - \hat{X}_h)^2 \tag{12}$$

  This metric penalizes large errors more heavily and emphasizes outlier sensitivity.

- **Mean Absolute Error (MAE)** MAE measures the average absolute deviation between predictions and ground truth:

$$\text{MAE} = \frac{1}{H} \sum_{h=1}^{H} \left| X_h - \hat{X}_h \right| \tag{13}$$

  It offers a more interpretable and scale-consistent view of typical prediction error.

## C.5 BASELINES

We compare our method against a broad set of baselines, covering foundation models, prompting-based approaches, context-driven forecasting architectures, and strong Transformer-based time series models. Below we briefly describe each baseline and its key modeling characteristics.

**TS Foundation Models:**

- **TabPFN** (Hoo et al., 2025): A transformer-based foundation model for tabular data trained on synthetic meta-tasks, capable of strong performance in low-data regimes. Though not specialized for time series, it offers insights into meta-learned inductive priors.

- **Chronos** (Ansari et al., 2024): A universal time series foundation model pretrained on large-scale multivariate data. It supports forecasting, classification, and imputation via a shared autoregressive decoder architecture.

- **MOIRAI** (Woo et al., 2024): A masked encoder-decoder transformer pre-trained on the LOTSA archive. It addresses universal forecasting by handling cross-frequency patterns, variable-length multivariate inputs, and distributional heterogeneity, thereby enabling cross-task generalization.

- **TEMPO** (Cao et al., 2023b): A GPT-style time series model that uses component-level decomposition (trend/seasonal/residual) and prompt-tuning. It supports zero-shot generalization across tasks using a unified autoregressive modeling framework.

- **GPT4TS** (Zhou et al., 2023): Adapts LLMs for time series forecasting via instruction tuning. It fine-tunes on forecasting, classification, and anomaly detection, achieving competitive results on diverse downstream tasks.

**LLM Prompting:**

- **GPT-4o**, **GPT-4o-mini** (Achiam et al., 2023), **LLaMA3-8B** (Touvron et al., 2023), **Mistral-7B** (Jiang et al., 2024): General-purpose LLMs prompted using natural language templates. These models encode both numerical series and contextual variables as text, offering a zero-shot baseline for forecasting with context understanding capabilities.

**Context-driven Approaches:**

- **TGForecaster** (Xu et al., 2024): A multimodal forecasting model that integrates textual information—such as dynamic news or channel descriptions—through cross-attention fusion. It is designed for the Text-Guided Time Series Forecasting (TGTSF) task and establishes strong baselines on multimodal benchmarks.

- **ContextIsKey** (Williams et al., 2024): Introduces a benchmark for integrating natural language context with numerical time series. It shows that LLM-based prompting methods can outperform conventional models by leveraging human-like contextual reasoning.

- **ContextMatters** (Chattopadhyay et al., 2024): Proposes ContextFormer, a plug-and-play module to inject heterogeneous and multimodal contextual information—categorical, continuous, or textual—into base forecasters. It significantly improves predictive performance through targeted context distillation.

- **TimeXer** (Wang et al., 2024b): A transformer architecture that explicitly models exogenous variables through patch-wise attention and cross-variate fusion. It is designed to reconcile endogenous dynamics and external signals for better scalability and accuracy.

**Transformer-based Approaches:**

- **PatchTST** (Nie et al., 2022): A Transformer-based model for multivariate time series forecasting that introduces two key designs: (i) segmentation of time series into subseries-level patches as input tokens, and (ii) channel-independent modeling where each univariate series shares weights across channels. This patching strategy preserves local semantics, reduces quadratic attention complexity, and enables longer effective history modeling, significantly improving long-term forecasting accuracy.

- **iTransformer** (Liu et al., 2023): Revisiting the Transformer design for time series, iTransformer inverts the typical input format by embedding time points into variate tokens. Attention is applied across variables rather than time steps, enabling better modeling of multivariate correlations without altering Transformer components. This design addresses common issues in long-horizon forecasting and improves interpretability by preserving variate-centric representations.

C.6 DETAILED COMPARISON WITH TIME-LLM, TTMS, AND TIMER-XL

This appendix elaborates on the uniqueness of our approach along two views: a high-level comparison of core philosophy and technical path, and a one-on-one detailed comparison with representative context-driven models.

C.6.1 CORE PHILOSOPHY AND TECHNICAL PATH (ONE-VS-MANY)

A high-level comparison highlights the fundamental distinction of our method.

**Core Focus.**

- **Our Work.** *Causality-driven context understanding*: focuses on the causal relationship between context and time series, aiming for interpretable and robust forecasts.
- **Time-LLM.** *LLM capability reprogramming*: applies LLMs to model time-series data.
- **TTMs.** *Model efficiency and generality*: targets a lightweight, fast, and generalizable model.
- **Timer-XL.** *Unified long-sequence forecasting*: enlarges context to capture long dependencies.

**Context Handling.**

- **Our Work.** *Causal attribution mechanism*: actively discovers and validates causal context via neural structural equation modules and counterfactual perturbations.
- **Time-LLM.** *Textual alignment*: converts data to text for LLM consumption.
- **TTMs.** *Signal fusion*: injects exogenous signals using an Exogenous Mixer.
- **Timer-XL.** *Long-dependency modeling*: captures dependencies with TimeAttention.

**Nature of Innovation.**

- **Our Work.** *Paradigm shift from correlation to causation*: transitions from using context merely as input to understanding its causal effect.
- **Time-LLM.** *Application paradigm innovation*: transfers NLP models to time series.
- **TTMs.** *Engineering/efficiency innovation*: optimizes architecture for speed and scale.
- **Timer-XL.** *Architectural and scale innovation*: enlarges receptive fields for broader modeling.

*Summary.* While other models primarily focus on better utilizing context, our approach centers on better understanding the *causal effect* of context.

C.6.2 ONE-ON-ONE DETAILED COMPARISON WITH EACH MODEL

**Comparison with Time-LLM: Addressing "Semantic Ambiguity" rather than "Modality Alignment".** Time-LLM emphasizes *modality alignment*, enabling LLMs to handle time series by reprogramming inputs into a text format. In contrast, our method targets *semantic-level causal* challenges:

- **Causal Selectivity.** Prefix prompting introduces context but cannot ensure attention to truly causal factors. Our design employs a *Neural Structural Equation Module* together with *Counterfactual Validation* to discover and confirm causal relations.
- **Heterogeneous Context.** Converting non-text data to text risks losing structure. A *Context-aware Attention Module* is used to dynamically handle both structured and unstructured data.

*Conclusion.* The goal is not to make LLMs merely read time series, but to enable understanding of the *causal logic* of context, thereby improving robustness.

**Comparison with TTMs: Pursuing "Deep Causal Explanation" rather than "Efficient Signal Fusion".** TTMs achieve high efficiency by fusing context via an *Exogenous Mixer*. Our perspective differs on two fronts:

- **Context Fusion.** Instead of direct fusion, a *Query Modulation Mechanism* guided by context semantics steers how contextual information influences forecasting queries.

- **Attribution.** Rather than relying on post-hoc explanations, *causal attribution* is built into the model and supported by entropy regularization for interpretability.

*Conclusion.* Some simplicity is traded for deeper *causal modeling* and interpretability, focusing not only on *what* context matters but also *why* and *how* it matters.

**Comparison with Timer-XL: Emphasizing "Causal Priors" rather than only "Enlarging the Receptive Field".** Timer-XL models long sequences and covariates with correlation-based attention. Our stance differs in:

- **Attention Driver.** Timer-XL relies on similarity-based attention, whereas attention is transformed into *causal inference* via inductive bias, enabling focus on key drivers beyond correlation.

- **Robustness.** Timer-XL performs well under stable distributions but degrades when correlations shift. Causal grounding in our approach supports stronger *out-of-distribution* generalization.

*Conclusion.* Rather than passively expecting the model to learn the correct dependencies, the method *actively injects causal structures* to uncover robust drivers.

### C.7 IMPLEMENTATION DETAILS

We follow the experimental configurations outlined in (Zhou et al., 2023) to ensure fair and reproducible evaluation across baselines https://github.com/thuml/Time-Series-Library. All models are implemented using PyTorch (Imambi et al., 2021), and all experiments are conducted on a single NVIDIA A100 GPU.

We adopt GPT2 (Radford et al., 2019) as our backbone model, enabling the first 6 hidden layers by default. Pretrained weights are loaded from HuggingFace Transformers (Wolf et al., 2020). Following common practice (Cao et al., 2023b), each sequence is segmented into non-overlapping patches with length $P = 16$ and stride $S = 8$. Dropout is applied with a fixed rate of 0.1 across all experiments. We set the learning rate to 0.0001 and adjust batch size according to dataset scale, selected from $\{256, 128, 64, 32, 16\}$. Early stopping is triggered after three epochs without improvement on the validation loss. Each experiment is repeated three times, and we report the averaged results.

We tune key loss coefficients via grid search: the counterfactual regularization weight $\lambda_{\text{CF}}$ is selected from $\{1.0, 0.1, 0.01\}$, and the entropy regularization weight $\lambda_{\text{ent}}$ is selected from $\{0.01, 0.001, 0.0001\}$. Unless otherwise stated, the loss function is Mean Squared Error (MSE). For causal context selection, we apply a top-$k$ mask with $k = 50\%$ of the total context dimensions retained, chosen for simplicity and computational efficiency. The structural equation module $\psi_{\text{SEM}}(\cdot)$ is implemented as a single-layer nonlinear neural network to model the relationship between context and target dynamics in a lightweight manner.

### C.8 EXPERIMENTAL RESULTS

#### C.8.1 MAIN RESULTS

Table 5 presents the full forecasting results (MSE/MAE) across all datasets, horizons, and baseline methods. These comprehensive comparisons supplement the main paper and highlight the consistent performance of our method.

#### C.8.2 CAUSAL DRIVERS ANALYSIS

To figure out how causal context contributes to prediction quality, we present two diagnostic cases that illustrate distinct modes of context influence (Figure 3). The first case involves Bitcoin price forecasting during a period of sharp market shift in May 2020. At this point, contextual signals such as rising market capitalization (from 1.44 to $1.60 \times 10^{13}$) and increasing Google Trends index (from 44 to 63) reflected heightened public attention and speculative activity. The context-aware model

Table 5: Full-shot forecasting performance across five datasets. Each cell shows MSE/MAE. Lower is better.

| Dataset | Horizon | Ours | TGForecaster | ContextMatters | TimeXer | Chronos | TEMPO | GPT4TS | PatchTST | iTransformer |
|---|---|---|---|---|---|---|---|---|---|---|
| Electricity | 96 | 0.2858/0.3739 | 0.3344/0.4168 | 0.2858/0.3772 | 0.2766/0.3699 | 0.3045/0.3861 | 0.3211/0.4070 | 0.3236/0.4050 | 0.2998/0.3860 | 0.2982/0.3889 |
| | 192 | 0.3018/0.3827 | 0.3808/0.4435 | 0.3267/0.4009 | 0.3182/0.3932 | 0.3283/0.4922 | 0.3324/0.4092 | 0.3360/0.4102 | 0.3344/0.4043 | 0.3414/0.4206 |
| | 336 | 0.3641/0.4216 | 0.4352/0.4753 | 0.3971/0.4412 | 0.3790/0.4263 | 0.3740/0.4287 | 0.3787/0.4397 | 0.3817/0.4397 | 0.3960/0.4408 | 0.4002/0.4511 |
| | 720 | 0.4504/0.4861 | 0.5253/0.5352 | 0.5127/0.5215 | 0.4699/0.4942 | 0.4536/0.4865 | 0.4499/0.4929 | 0.4532/0.4948 | 0.4905/0.5100 | 0.4908/0.5110 |
| | avg | 0.3505/0.4161 | 0.4189/0.4677 | 0.3805/0.4352 | 0.3609/0.4209 | 0.3651/0.4484 | 0.3705/0.4361 | 0.3737/0.4374 | 0.3802/0.4353 | 0.3827/0.4429 |
| Exchange | 96 | 0.1017/0.2341 | 0.0977/0.2337 | 0.1112/0.2439 | 0.1029/0.2363 | 0.1037/0.2383 | 0.1076/0.2418 | 0.1017/0.2357 | 0.1116/0.2495 | 0.1278/0.2536 |
| | 192 | 0.2025/0.3347 | 0.2146/0.3417 | 0.2154/0.3502 | 0.2100/0.3423 | 0.2144/0.3492 | 0.2123/0.3453 | 0.2114/0.3441 | 0.2349/0.3683 | 0.2393/0.3605 |
| | 336 | 0.4172/0.4840 | 0.4407/0.4860 | 0.4244/0.4887 | 0.4406/0.4979 | 0.4182/0.4915 | 0.4094/0.4839 | 0.4244/0.4915 | 0.4200/0.4913 | 0.4327/0.4944 |
| | 720 | 1.0826/0.7924 | 1.1032/0.8087 | 1.1102/0.8004 | 1.0653/0.7864 | 1.1058/0.8074 | 1.1445/0.8162 | 1.1480/0.8241 | 1.1305/0.8103 | 1.1404/0.8323 |
| | avg | 0.4510/0.4613 | 0.4641/0.4675 | 0.4653/0.4708 | 0.4547/0.4657 | 0.4605/0.4716 | 0.4684/0.4718 | 0.4714/0.4739 | 0.4743/0.4799 | 0.4851/0.4852 |
| Bitcoin | 36 | 0.0979/0.2322 | 0.1024/0.2415 | 0.0987/0.2366 | 0.1045/0.2436 | 0.1132/0.2539 | 0.1309/0.2786 | 0.1459/0.3014 | 0.1184/0.2565 | 0.1177/0.2546 |
| | 36 | 0.1410/0.2842 | 0.1512/0.3017 | 0.1493/0.2977 | 0.1521/0.3025 | 0.1605/0.3087 | 0.1666/0.3101 | 0.1730/0.3320 | 0.1744/0.3365 | 0.1737/0.3366 |
| | 48 | 0.1725/0.3167 | 0.1847/0.3295 | 0.1810/0.3314 | 0.1844/0.3333 | 0.1883/0.3405 | 0.1893/0.3459 | 0.1896/0.3452 | 0.1929/0.3580 | 0.1944/0.3522 |
| | 60 | 0.1962/0.3490 | 0.2090/0.3579 | 0.2158/0.3575 | 0.2133/0.3613 | 0.1973/0.3741 | 0.1938/0.3706 | 0.2008/0.3734 | 0.2281/0.3754 | 0.2334/0.3849 |
| | avg | 0.1519/0.2955 | 0.1618/0.3077 | 0.1612/0.3058 | 0.1636/0.3102 | 0.1648/0.3193 | 0.1701/0.3263 | 0.1773/0.3380 | 0.1784/0.3316 | 0.1798/0.3321 |
| ILI | 36 | 0.6554/0.6303 | 0.7521/0.6498 | 0.7746/0.6776 | 0.6632/0.6354 | 0.8374/0.7153 | 0.9825/0.8129 | 0.9766/0.8262 | 0.8074/0.6939 | 0.8515/0.7065 |
| | 36 | 0.6802/0.6653 | 0.7374/0.6783 | 0.8388/0.7470 | 0.7026/0.6775 | 0.8529/0.7367 | 0.9091/0.7850 | 0.9675/0.8182 | 0.8803/0.7726 | 0.8864/0.7809 |
| | 48 | 0.7674/0.7319 | 0.7573/0.6730 | 0.8584/0.7787 | 0.7495/0.7218 | 0.9128/0.7952 | 0.9961/0.8254 | 1.0590/0.8613 | 0.8903/0.7969 | 0.8988/0.7991 |
| | 60 | 0.7792/0.7394 | 0.7807/0.7102 | 0.9649/0.8456 | 0.7981/0.7456 | 0.9663/0.8474 | 1.0863/0.8714 | 1.2305/0.9373 | 0.9821/0.8545 | 0.9564/0.8128 |
| | avg | 0.7205/0.6917 | 0.7585/0.6872 | 0.8874/0.7905 | 0.7501/0.7150 | 0.9107/0.7931 | 0.9972/0.8272 | 1.0857/0.8723 | 0.9175/0.8080 | 0.9138/0.7976 |
| Traffic | 96 | 0.1905/0.2768 | 0.2803/0.3688 | 0.1919/0.2798 | 0.1958/0.2928 | 0.2335/0.3319 | 0.2411/0.3409 | 0.2559/0.3516 | 0.2369/0.3345 | 0.2423/0.3427 |
| | 192 | 0.1914/0.2809 | 0.2762/0.3671 | 0.1969/0.2889 | 0.2004/0.2952 | 0.2278/0.3276 | 0.2162/0.3195 | 0.2239/0.3231 | 0.2386/0.3332 | 0.2390/0.3414 |
| | 336 | 0.1835/0.2768 | 0.2692/0.3668 | 0.1872/0.2787 | 0.1975/0.2942 | 0.2036/0.3142 | 0.2125/0.3210 | 0.2121/0.3129 | 0.2428/0.3406 | 0.2452/0.3514 |
| | 720 | 0.2139/0.3035 | 0.2804/0.3742 | 0.2175/0.3082 | 0.2199/0.3131 | 0.2261/0.3287 | 0.2282/0.3325 | 0.2388/0.3373 | 0.2595/0.3676 | 0.2670/0.3692 |
| | avg | 0.1948/0.2845 | 0.2765/0.3692 | 0.1984/0.2889 | 0.2034/0.2991 | 0.2228/0.3256 | 0.2245/0.3285 | 0.2327/0.3313 | 0.2444/0.3440 | 0.2484/0.3512 |
| ETTh1 | 96 | 0.0540/0.1766 | 0.0591/0.1856 | 0.0577/0.1843 | 0.0568/0.1808 | 0.0581/0.1839 | 0.0563/0.1809 | 0.0564/0.1800 | 0.0576/0.1825 | 0.0571/0.1826 |
| | 192 | 0.0713/0.2013 | 0.0753/0.2096 | 0.0728/0.2062 | 0.0735/0.2070 | 0.0733/0.2068 | 0.0729/0.2066 | 0.0738/0.2077 | 0.0754/0.2104 | 0.0730/0.2066 |
| | 336 | 0.0816/0.2216 | 0.0895/0.2342 | 0.0924/0.2374 | 0.0862/0.2300 | 0.0827/0.2265 | 0.0856/0.2293 | 0.0874/0.2318 | 0.0904/0.2346 | 0.0957/0.2384 |
| | 720 | 0.0926/0.2378 | 0.1005/0.2484 | 0.1039/0.2541 | 0.0959/0.2439 | 0.0936/0.2403 | 0.0946/0.2458 | 0.0925/0.2399 | 0.0986/0.2475 | 0.0970/0.2331 |
| | avg | 0.0749/0.2093 | 0.0811/0.2194 | 0.0817/0.2205 | 0.0781/0.2154 | 0.0769/0.2144 | 0.0773/0.2156 | 0.0775/0.2148 | 0.0805/0.2188 | 0.0807/0.2152 |
| ETTh2 | 96 | 0.1358/0.2847 | 0.1386/0.2888 | 0.1468/0.2964 | 0.1384/0.2857 | 0.1352/0.2771 | 0.1327/0.2789 | 0.1305/0.2766 | 0.1425/0.2906 | 0.1433/0.2937 |
| | 192 | 0.1850/0.3388 | 0.1893/0.3428 | 0.1914/0.3441 | 0.1882/0.3406 | 0.1809/0.3327 | 0.1830/0.3340 | 0.1793/0.3301 | 0.1939/0.3457 | 0.1843/0.3380 |
| | 336 | 0.2135/0.3704 | 0.2207/0.3751 | 0.2355/0.3863 | 0.2183/0.3730 | 0.2188/0.3736 | 0.2199/0.3739 | 0.2204/0.3740 | 0.2277/0.3805 | 0.2190/0.3746 |
| | 720 | 0.2268/0.3819 | 0.2335/0.3880 | 0.2362/0.3892 | 0.2334/0.3875 | 0.2385/0.3924 | 0.2441/0.3974 | 0.2472/0.4002 | 0.2349/0.3886 | 0.2382/0.3917 |
| | avg | 0.1903/0.3439 | 0.1955/0.3487 | 0.2025/0.3540 | 0.1946/0.3467 | 0.1934/0.3440 | 0.1949/0.3461 | 0.1943/0.3452 | 0.1997/0.3513 | 0.1962/0.3495 |
| ETTm1 | 96 | 0.0288/0.1259 | 0.0306/0.1308 | 0.0306/0.1305 | 0.0294/0.1276 | 0.0303/0.1302 | 0.0289/0.1265 | 0.0295/0.1275 | 0.0307/0.1306 | 0.0288/0.1273 |
| | 192 | 0.0433/0.1579 | 0.0467/0.1648 | 0.0446/0.1600 | 0.0444/0.1600 | 0.0440/0.1586 | 0.0437/0.1590 | 0.0443/0.1598 | 0.0469/0.1647 | 0.0460/0.1639 |
| | 336 | 0.0585/0.1837 | 0.0608/0.1906 | 0.0638/0.1925 | 0.0591/0.1852 | 0.0583/0.1849 | 0.0577/0.1845 | 0.0581/0.1851 | 0.0619/0.1904 | 0.0619/0.1909 |
| | 720 | 0.0807/0.2175 | 0.0849/0.2246 | 0.0855/0.2243 | 0.0833/0.2211 | 0.0831/0.2195 | 0.0820/0.2184 | 0.0828/0.2188 | 0.0851/0.2240 | 0.0810/0.2188 |
| | avg | 0.0528/0.1713 | 0.0558/0.1777 | 0.0561/0.1768 | 0.0540/0.1735 | 0.0539/0.1733 | 0.0531/0.1721 | 0.0537/0.1728 | 0.0562/0.1774 | 0.0544/0.1752 |
| ETTm2 | 96 | 0.0655/0.1838 | 0.0720/0.1955 | 0.0659/0.1838 | 0.0636/0.1826 | 0.0645/0.1852 | 0.0681/0.1861 | 0.0690/0.1877 | 0.0679/0.1844 | 0.0692/0.1895 |
| | 192 | 0.1007/0.2358 | 0.1045/0.2419 | 0.0999/0.2339 | 0.0990/0.2347 | 0.1004/0.2363 | 0.1010/0.2354 | 0.1011/0.2350 | 0.1023/0.2359 | 0.1057/0.2428 |
| | 336 | 0.1285/0.2721 | 0.1318/0.2773 | 0.1342/0.2759 | 0.1331/0.2776 | 0.1312/0.2756 | 0.1303/0.2745 | 0.1292/0.2768 | 0.1307/0.2727 | 0.1396/0.2858 |
| | 720 | 0.1793/0.3268 | 0.1868/0.3358 | 0.1875/0.3357 | 0.1848/0.3332 | 0.1854/0.3317 | 0.1841/0.3326 | 0.1831/0.3296 | 0.1836/0.3313 | 0.1848/0.3367 |
| | avg | 0.1185/0.2546 | 0.1238/0.2626 | 0.1219/0.2573 | 0.1201/0.2570 | 0.1204/0.2572 | 0.1209/0.2571 | 0.1206/0.2573 | 0.1211/0.2566 | 0.1249/0.2637 |
| Weather | 96 | 0.0013/0.0270 | 0.0016/0.0291 | 0.0013/0.0259 | 0.0013/0.0265 | 0.0013/0.0261 | 0.0013/0.0265 | 0.0013/0.0263 | 0.0013/0.0274 | 0.0013/0.0282 |
| | 192 | 0.0014/0.0268 | 0.0015/0.0288 | 0.0015/0.0285 | 0.0016/0.0297 | 0.0016/0.0292 | 0.0016/0.0300 | 0.0016/0.0294 | 0.0016/0.0302 | 0.0016/0.0314 |
| | 336 | 0.0016/0.0298 | 0.0018/0.0315 | 0.0017/0.0307 | 0.0017/0.0315 | 0.0017/0.0315 | 0.0017/0.0316 | 0.0017/0.0312 | 0.0017/0.0323 | 0.0018/0.0335 |
| | 720 | 0.0021/0.0344 | 0.0020/0.0328 | 0.0021/0.0346 | 0.0022/0.0356 | 0.0022/0.0361 | 0.0022/0.0361 | 0.0022/0.0358 | 0.0021/0.0363 | 0.0022/0.0364 |
| | avg | 0.0016/0.0295 | 0.0017/0.0306 | 0.0017/0.0299 | 0.0017/0.0308 | 0.0017/0.0307 | 0.0017/0.0311 | 0.0017/0.0307 | 0.0017/0.0316 | 0.0017/0.0324 |
| BJAQ | 96 | 1.2214/0.7490 | 1.3967/0.7709 | 1.5183/0.8180 | 1.2140/0.7406 | 1.2265/0.7458 | 1.2725/0.7545 | 1.2139/0.7423 | 1.3279/0.7622 | 1.3361/0.7657 |
| | 192 | 1.3298/0.7755 | 1.7071/0.8561 | 1.5578/0.8496 | 1.3350/0.7826 | 1.3416/0.7922 | 1.3558/0.7956 | 1.3438/0.7981 | 1.3804/0.8029 | 1.3976/0.7927 |
| | 336 | 1.3957/0.8089 | 1.8655/0.8997 | 1.6519/0.8717 | 1.3925/0.8048 | 1.3884/0.8077 | 1.3861/0.8090 | 1.3814/0.8069 | 1.4717/0.8264 | 1.4170/0.8070 |
| | 720 | 1.3837/0.7983 | 1.5150/0.8364 | 1.5090/0.8505 | 1.4171/0.8187 | 1.4063/0.8152 | 1.4022/0.8091 | 1.4240/0.8215 | 1.4367/0.8243 | 1.4781/0.8239 |
| | avg | 1.3327/0.7829 | 1.6211/0.8407 | 1.5593/0.8474 | 1.3396/0.7867 | 1.3407/0.7902 | 1.3542/0.7921 | 1.3408/0.7922 | 1.4042/0.8040 | 1.4072/0.7973 |

captures the upward trend by incorporating macro-level indicators as causal drivers in its prediction. In contrast, the model without context follows only temporal trends and misses the structural change, resulting in a flat forecast. This highlights the importance of causal context in providing early signals that precede and help explain shifts in the prediction target.

The second case focuses on influenza prediction using the ILI dataset, where the overall case count transitions from decline to stabilization. Structured internal context includes age-specific infection rates, which serve as fine-grained signals for transmission dynamics. Among these, the 5–10 age group exhibits a rise–peak–plateau pattern that precedes and aligns with the turning point in total cases. This subgroup likely plays a key role in community spread due to high contact rates in schools. The model with context captures this temporal coupling and accurately predicts the stabilization phase. In contrast, the model without context extrapolates the declining trend, failing to detect the causal shift reflected in subgroup dynamics. This example illustrates how internal context can act as a high-resolution early indicator when structurally linked to the generative process of the target.

### C.8.3 Ablation Study

We conduct ablation experiments to evaluate the role of each component in our framework. Results are reported in Table 6. Removing all contextual input (**w/o Context**) leads to the most severe performance drop. This confirms that temporal modeling alone is insufficient, and that context plays a key role in many real-world settings. We next examine a baseline that uses plain cross-attention over context without any reasoning mechanism (**r/p Vanilla Attention**). Compared to our full model, this

variant shows consistent degradation. Although this variant uses contextual input, it lacks mechanisms to assess the causal influence of each signal. This highlights that reliable context integration requires not just access, but principled reasoning over causal relevance. We further evaluate the impact of removing each structural enhancement in isolation. Excluding the causal bias module (**w/o Causal Bias**) results in the largest drop among the three. This suggests that assigning structural importance to context dimensions is essential for generalization, particularly under variable or noisy inputs. Removing query modulation (**w/o Query Modulation**) also leads to weaker performance, especially on Bitcoin and ILI. This shows that aligning temporal queries with global context semantics helps improve attention selectivity. Finally, removing the counterfactual loss (**w/o Counterfactual Loss**) produces a moderate decline in accuracy, particularly on datasets with unstable context-target relationship. This supervision helps the model distinguish useful features by enforcing contrast between factual and counterfactual outcomes. Overall, the results support the design of causal-aware modules for effective context integration.

Table 6: Ablation study results across five datasets. Each row removes a specific component of the proposed framework. Lower MSE and MAE indicate better performance.

| Model | Bitcoin | | ILI | | Weather | | Electricity | | BJAQ | |
|---|---|---|---|---|---|---|---|---|---|---|
| | MSE | MAE | MSE | MAE | MSE | MAE | MSE | MAE | MSE | MAE |
| **Ours** | **0.1519** | **0.2955** | **0.7223** | **0.6928** | **0.0016** | **0.0295** | **0.3505** | **0.4161** | **1.3338** | **0.7835** |
| w/o Counterfactual Loss | 0.1542 | 0.2974 | 0.7287 | 0.6993 | 0.0017 | 0.0301 | 0.3575 | 0.4176 | 1.3375 | 0.7881 |
| w/o Query Modulation | 0.1553 | 0.2987 | 0.7275 | 0.6984 | 0.0017 | 0.0298 | 0.3524 | 0.4173 | 1.3349 | 0.7850 |
| w/o Causal Bias | 0.1581 | 0.3006 | 0.7251 | 0.6933 | 0.0017 | 0.0297 | 0.3531 | 0.4174 | 1.3362 | 0.7855 |
| w/o Context | 0.1767 | 0.3368 | 1.0537 | 0.8661 | 0.0017 | 0.0312 | 0.3715 | 0.4358 | 1.3417 | 0.7905 |
| r/p Vanilla Attention | 0.1632 | 0.3076 | 0.7537 | 0.7261 | 0.0017 | 0.0305 | 0.3676 | 0.4258 | 1.3394 | 0.7893 |

# D   RELATED WORK

## D.1   CONTEXT-AWARE TIME SERIES FORECASTING

Incorporating auxiliary signals has long been central to improving time series forecasting. Classical methods such as ARIMAX Contreras et al. (2003) and SARIMAX (Dubey et al., 2021) explicitly account for exogenous variables through autoregressive formulations. With the rise of deep learning, models like DeepAR (Salinas et al., 2020), TFT (Lim et al., 2021), and NBEATSx (Olivares et al., 2023) integrate covariates via concatenation or attention-based selection. More recent architectures such as TiDE (Das et al., 2023), TimeXer (Wang et al., 2024b), and TSMixer (Chen et al., 2023) leverage patch-level or metadata-aware mechanisms to encode temporal signals more flexibly. Despite their success, these methods often assume that all contextual inputs are uniformly relevant, lacking explicit filtering mechanisms and leaving models vulnerable to spurious correlations—especially under distribution shifts. Recent work extends the notion of context beyond structured covariates to include multimodal and textual sources (Xu et al., 2024; Hong et al., 2025). The CONTEXT IS KEY benchmark (Williams et al., 2024) categorizes context into causal, future-aware, and textual forms, and evaluates model performance on tasks where such signals are critical. Methods like ContextMatters (Chattopadhyay et al., 2024) and TimeXer incorporate heterogeneous metadata via cross-attention or hierarchical encoders, but treat contextual variables indiscriminately. Several LLM-based approaches, including ChatTime (Wang et al., 2025), GPT4MTS (Jia et al., 2024), and CALF (Liu et al., 2025b), explore textual context integration through prompt learning and modality alignment. While promising in zero-shot settings, these techniques primarily target semantic fusion and lack mechanisms to identify which signals are actually predictive. This highlights an emerging need for forecasting frameworks that can unify heterogeneous context while distinguishing signal from noise in a principled manner. Going beyond static fusion or textual prompting, such methods should incorporate dynamic context selection, causal interpretability, and architectural flexibility—especially under settings that demand transferability across domains or data modalities.

## D.2   LLM-DRIVEN FOUNDATION MODELS FOR TIME SERIES FORECASTING

Transformer-based architectures have become a cornerstone in modern time series forecasting, offering strong scalability and capacity to model long-range dependencies (Zhou et al., 2021; Liu et al.,

2022; Chen et al., 2024b; Zhang & Yan, 2023) . Early designs such as iTransformer (Liu et al., 2023), Autoformer (Wu et al., 2021), and PatchTST (Nie et al., 2022) encode complex temporal patterns via sparse attention or patch-based tokenization, forming a robust foundation for sequence modeling. Building on this architectural backbone, recent work explores the use of large language models (LLMs) for time series tasks—leveraging their powerful sequence reasoning capabilities and growing pretraining paradigms (Liu et al., 2025a). A prominent line of work treats numerical series as text, enabling zero-shot forecasting through prompt-based generation. For example, LLMTIME reformulates time series as digit sequences and applies GPT-style models for direct next-token prediction, revealing LLMs' inductive bias toward periodicity and simplicity. While intuitive, such methods suffer from inefficient sampling and limited scalability. TIME-LLM (Jin et al., 2023) and GPT4TS (Zhou et al., 2023) attempt to improve alignment by introducing prototype-conditioned prompts or lightweight adaptation of frozen backbones, yet still require in-domain tuning. TEMPO (Cao et al., 2023b) takes a step further by decomposing time series into trend-seasonal components and injecting soft prompts to improve generalization, but relies on manually designed inductive structures. Similarly, UNITIME (Liu et al., 2024b) augments forecasting with domain-specific instructions using a language-time series transformer, demonstrating the benefit of cross-modal alignment, though still constrained by domain labeling and convergence imbalances. A complementary direction constructs foundation models trained directly on time series, avoiding dependence on textual pretraining. CHRONOS (Ansari et al., 2024) pioneers this approach by treating quantized time series as a new language and training transformers from scratch, achieving strong zero-shot generalization. MOIRAI (Woo et al., 2024) extends this idea with a unified masked-encoder design that handles any-variate inputs and heterogeneous distributions, while TIMEGPT (Garza et al., 2023), LAG-LLAMA (Rasul et al., 2023), and TIMESFM (Das et al., 2024) further scale up pretraining on diverse datasets for general-purpose forecasting. However, most of these models still treat time series as isolated numeric signals, overlooking the contextual semantics or covariates crucial for real-world deployment. Our work builds on this foundation by proposing a context-enhanced time series model, which integrates external knowledge into the pretraining process to better capture latent temporal structures and improve robustness under domain shifts.

### D.3 CAUSAL MODELING IN TEMPORAL SETTINGS

Temporal causal modeling began in epidemiology and is often based on G-computation, marginal structural models (MSMs), and structural nested models Robins (1986); Robins et al. (2000); Robins & Hernan (2008). Early methods relied on linear assumptions and struggled to capture complex temporal patterns. Later works improved expressiveness by using Bayesian nonparametrics (Xu et al., 2016; Soleimani et al., 2017; Schulam & Saria, 2017) and deep models such as RMSNs (Lim, 2018) and G-Net (Li et al., 2021), which apply RNN-based architectures to forecast treatment outcomes. Recent approaches aim to learn representations that are both predictive and invariant to treatment (Ganin et al., 2016; Tzeng et al., 2015). Examples include CRN (Bica et al., 2020) and Transformer-based models (Melnychuk et al., 2022; Vaswani et al., 2017), which often use joint losses or domain confusion. Some studies extend counterfactual reasoning to irregular time series (Seedat et al., 2022; Cao et al., 2023a), point processes (Zhang et al., 2022), and spatiotemporal graphs (Jiang et al., 2023; Kidger et al., 2020). Although many methods focus on balancing, recent work shows that expressiveness plays a more important role in robust performance Meng et al. (2023). To handle unmeasured confounders, some models use proxies or frequency-based decomposition Cao et al. (2023a), but these often require strong assumptions and may not generalize well.

## E DECLARATION ON THE USE OF LARGE LANGUAGE MODELS

Portions of this manuscript were refined with the assistance of GPT-5 to improve readability, structure, and grammar. The study's conception, methodology, analyses, and conclusions are entirely the authors' work. All AI-assisted edits were reviewed by the authors, who assume full responsibility for the final text.

