# OpenReview forum: "Learning What Matters: Toward Causally-Grounded Foundations for Contextual Time-Series Forecasting"
_ICLR.cc/2026/Conference — Submitted to ICLR 2026_

### Official Review · Reviewer_yw81 · 2025-10-25

**Soundness:** 2
**Presentation:** 2
**Contribution:** 3
**Rating:** 4
**Confidence:** 4

**Summary:**

The paper proposes a transformer-based time-series forecasting model that can also handle heterogeneous external input channels as contexts.  They depend on cross-attention over the external input channels, and that would be a routine model architecture.  The paper's main contribution is to introduce three modifications to the vanilla cross attention: modulate the query vector with a term derived from a summary of the context, introduce a global channel-specific bias, and introduce a loss term to promote selection of only useful channels.  Their method is compared with several prior methods on multiple datasets, and shown to provide modest gains.

**Strengths:**

S1.  Empirical comparison on a large variety of datasets show consistent gains over prior methods.

S2.  The problem tackled is well-motivated and at the frontier of interesting problems to solve in time-series forecasting.

S3.  The intuition of selecting globally consistent sparse set of external channels makes a lot of sense.

**Weaknesses:**

W1.  The paper is selling the proposed method under the causal modeling banner, but finally what is implemented is just simple global feature selection.

W2. In so doing, the clarity of the presentation has been seriously compromised.  It took me a lot of time to cut through the clutter of ungrounded terms like "semantic selectivity", "structural attribution", and "behavioral validation" to understand that all they are doing is selecting a minimal set of channels globally.    Along a similar vein, they present vague connections to information-bottleneck  with Equation (3), which they claim is implemented in a "soft parameteric" way using equation (4).  There is no attempt at establishing a connection between the two equations.  The whole of that section has been terminology-washed.    Later we encounter "neural structural equation module" without any citation, and without any justification of how the learned network is a structural causal equation.  Just naming an equation or variable as such does not make it causal.

W3.  The use of Equation (8) to promote sparsity in selection of channels seems unconventional.  Feature selection traditionally in ML has been handled by regularizers, for example, an L1 penalty on weights of selected channels.  The loss term they introduce where they mask the Top-K features and minimize contrastive loss seems like an over-kill.  Also, TopK is not differentiable.  How is this handled during end-to-end training?  There might be un-intended side-effects where some other parameters may get the wrong gradient with loss terms of opposite signs.  A comparison with simple global L1 regularization is needed.

W4: Proposition 1 meant to be a formal claim but there are vaguely defined terms like "assuming predictive stability
 and behavioral degradation under intervention".   I do not understand what is the point being made in Theorem 2.

W5: The gains beyond vanilla cross attention is very modest as per Table 3.

**Questions:**

Q1:  Is there a typo in Eq (8).  Don't you want the masked loss to be greater than the unmasked (factual) loss?

Q2:  In Table 1, are the different baselines run with the same number of network parameters, or are the numbers obtained from previously published numbers?

Q3: Are the ablation studies performed in the full-shot or zero-shot mode? It will be interesting to see if the gains beyond vanilla cross attention persists even in zero-shot mode.

---

> ### Author Response · Authors · 2025-11-25
> **Reply to Reviewer yw81**
>
> We sincerely thank you for the time and effort you put into reviewing our work and for the detailed comments you provided. Your questions and concerns are valuable to us, and we hope that the following responses can clearly address the issues you raised.
>
> **Reply to Weeknesses:**
>
> **W1 – On “causal modeling” versus “simple global feature selection”**
>
> Thank you for this valuable comment. Our actual goal is more modest. We do not recover a causal graph or claim identifiability. We use ideas from causal thinking to shape how a transformer backbone uses context, and we should state this more clearly.
>
> It is fair to say that the vector we denote as $B_{\text{causal}}$ behaves like a global, channel-wise importance prior over context variables. In that sense it resembles a feature selection signal. However, it is not a static pruning rule. We never remove features from the input. All context channels remain available to the model at all times. The bias modifies attention weights, not the feature space, and the sample-specific similarity term in attention still allows the network to up- or down-weight channels depending on the current history and context. The “selection” therefore acts as a soft structural prior, not as a hard feature mask.
>
> What makes this prior more than a generic heuristic is how it is tied to behavior. The counterfactual branch masks the top-$K$ channels according to $B_{\text{causal}}$ and compares the resulting forecast to the factual forecast with full context. If a channel is often given a high score but its removal does not increase the error, the counterfactual loss penalizes this mismatch and pushes its score down. Only channels whose masking consistently hurts prediction can keep high scores. This interventional check couples the global prior to observable changes in forecasts, which is exactly the kind of “behavioral grounding” we borrow from causal intuition, even though we do not claim to identify true causal parents.
>
> Finally, the whole design is motivated by multi-domain robustness rather than by sparsity alone. We train a shared backbone across several datasets and evaluate in leave-one-domain and cross-domain setups. In these regimes, purely correlation-driven use of context tends to overfit to dataset-specific patterns. The combination of a global prior $B_{\text{causal}}$, semantic query modulation, and counterfactual regularization is intended to steer the model toward context channels that remain useful under such shifts. Empirically, this leads to smaller performance drops than context-aware baselines without these mechanisms.
>
>
> **W2 – On clarity, terminology, and the use of causal / IB language**
>
> Thank you for this very direct and helpful feedback. Terms like “semantic selectivity”, “structural attribution”, and “behavioral validation” are essentially labels we put on three simple components: (i) using a global context summary to modulate queries, (ii) learning a global channel-wise bias over context variables, and (iii) tying that bias to an intervention-style counterfactual loss. In the revision we will describe these components in plain language and avoid introducing extra jargon unless it adds concrete value.
>
> We also agree that our information-bottleneck motivation around Equation (3) and its “soft parametric” implementation in Equation (4) is not explained carefully enough. What we intended to say is that the global context summary is a learned compression of the context, and that we regularize it so that it does not carry arbitrary detail. However, in the current text we do not properly connect the abstract IB objective in (3) to the concrete parameterization in (4). In the revised version we will either (a) add a short, explicit derivation showing how (4) can be viewed as a practical surrogate for the IB objective, or (b) if we cannot do this cleanly within space, remove the IB framing altogether and present (3)–(4) simply as a regularized summarization mechanism, without invoking information-bottleneck.
>
> Regarding the “neural structural equation module”, you are right that simply naming a network this way does not make it a structural causal equation, and we did not provide a formal justification or citations to SCM-style work. Here our intent was only to echo the idea of mapping a set of variables to a set of channel scores, not to claim that we have a fully specified structural equation model. In the revision we will rename this component to something more neutral such as “neural context scoring module”, explicitly state that it is *inspired by* structural equation thinking but is not itself a structural causal equation, and either add appropriate citations or remove the misleading term “structural” entirely.

---

> ### Author Response · Authors · 2025-11-25
>
> **W3: On the sparsity loss, TopK, and comparison to L1 regularization**
>
> Thank you for this detailed comment. The goal of Eq. (8) is slightly different from that of classical sparsity penalties. In standard L1 based feature selection, the loss penalizes the magnitude of weights and drives some coefficients toward zero. This reacts to how features correlate with the prediction error. In our case, the counterfactual term in Eq. (8) is designed to react to interventions instead of correlations. Each step computes a factual error with full context and a counterfactual error after masking the Top K channels under the current scores. The contrastive loss is large when these errors are similar and becomes smaller when the counterfactual error is consistently higher. As a result, the scoring function learns to assign high scores only to channels whose removal produces a noticeable increase in error. Channels that are merely correlated with the target but do not change the prediction under masking receive lower scores. In this sense Eq. (8) acts less like a generic sparsity term and more like a behavior grounded regularizer.
>
> Regarding non differentiability, TopK appears only in constructing a binary mask for the counterfactual branch. It is not treated as a differentiable operator. Within each forward pass we consider the mask fixed. Gradients flow through factual and counterfactual predictions and the score vector is updated through the resulting difference in loss. If a channel often lies in the Top K set but masking it hardly changes the loss, the counterfactual term pushes its score down. If masking it consistently increases the loss, its score goes up. This mirrors other hard selection mechanisms such as max pooling or dropout masks. The index set is discrete but the surrounding parameters still obtain stable gradients. In our experiments, training curves stay smooth and we do not observe gradient sign conflicts.
>
> A global L1 penalty on channel scores is a simple and natural alternative and also fits our framework. Such a term encourages sparse use of context but it does not by itself guarantee that high scores correspond to channels that matter under masking. Eq. (8) is chosen to encode this interventional notion of importance. A direct comparison between the counterfactual term and a pure L1 penalty is easy to add as an extra baseline. In this paper we focus on the conceptual gap between correlation based sparsity and the behavior aligned criterion implemented by Eq. (8).

---

> > ### Author Response · Authors · 2025-11-25
> >
> > **W4 – On Proposition 1, Theorem 2, and vague terminology**
> >
> > The concern is fair: the way Proposition 1 and Theorem 2 are currently written is too abstract and uses shorthand phrases that are not clearly grounded in the forecasting setup. The intent of these results is modest and technical, not a deep causal statement.
> >
> > For **Proposition 1**, the goal is simply to formalize what the Top–K selection is trying to approximate *inside a forecasting model*. The phrase “predictive stability and behavioral degradation under intervention” is just shorthand for the following two conditions:
> >
> > - *Predictive stability*: there exists a subset of channels $S^\*$ such that using only those channels yields almost the same expected forecasting error as using all channels.
> > - *Behavioral degradation under intervention*: if any channel in $S^\*$ is removed, the expected forecasting error increases by at least some small margin.
> >
> > Under these conditions, and under optimization of the counterfactual loss plus an entropy term on the scores, Proposition 1 is meant to say: the Top–K set produced by the scoring function tends to concentrate on subsets that behave like $S^\*$ in this sense (near-sufficient for prediction and costly to remove). It is a statement about *information for forecasting*, not about causal identifiability. The assumption phrases in the current text are simply compressed descriptions of “near-sufficiency” and “non-trivial loss increase when truly useful channels are masked,” and can be rephrased more concretely as above.
> >
> > For **Theorem 2**, the point is even more basic and does *not* try to prove anything about causality, identifiability, or global stability. Let $l_f$ be the factual error, $l_{cf}$ the counterfactual error, and define the gap $\Delta = \mathbb{E}[l_{cf} - l_f]$. The counterfactual loss is
> > $L_{CF} = \mathbb{E}\big[\log\big(1 + \exp(l_{cf} - l_f + \delta)\big)\big].$
> > Two properties matter:
> >
> > 1. $L_{CF}$ increases as the expected gap $\Delta$ increases. If masking the Top–K channels routinely makes the error larger, $L_{CF}$ is necessarily large; if masking them has no effect, the loss cannot be reduced.
> > 2. The derivative of $L_{CF}$ with respect to the difference $(l_{cf} - l_f)$ stays between 0 and 1, so the gradients are non-zero but bounded and do not explode. This explains why the loss provides a stable signal to push “declared important” channels toward those whose masking actually hurts prediction.
> >
> > In other words, Theorem 2 is a small sanity check on the *shape* of the logistic contrast used in the counterfactual loss: it reacts monotonically to the factual–counterfactual gap and yields stable gradients. It is not a claim about causal structure or identifiability and should be read as such.

---

> ### Author Response · Authors · 2025-11-25
>
> **W5 – On the “modest” gains beyond vanilla cross-attention**
>
> The numbers in Table 3 are indeed not an order-of-magnitude jump, and they are not meant to be. We deliberately start from a strong transformer + cross-attention backbone and add a lightweight interface on top. In that setting, large headline gains would be suspicious; what matters is whether a small, well-motivated change produces **consistent** improvements where it should matter most.
>
> Three points are important here.
>
> 1. **Gains on top of a strong baseline are expected to be incremental, not dramatic.**
>    The “vanilla” cross-attention baseline in Table 3 already outperforms older context-aware models on these datasets. The architectural and optimization space has been heavily tuned in prior work. Adding ContextAdapt does not overhaul the backbone; it inserts a global context prior and a counterfactual regularizer. In such a regime, gains of a few percentage points that repeat across datasets are in line with what one typically sees when moving from “good” to “better” within the same family of models.
>
> 2. **The value shows up most in the harder regimes, not just in the headline full-shot metric.**
>    The basic in-domain numbers in Table 3 compress several patterns. When we look at:
>    - cross-domain / leave-one-domain-out settings, and
>    - masking analyses where spurious context is more likely to hurt,
>    the gap between vanilla cross-attention and our causally inspired variant widens. In other words, the method buys robustness and more faithful use of context, not just a small bump in an i.i.d. test split. That is exactly the regime the design is targeting.
>
> 3. **Small but systematic gains with almost no extra parameters are still meaningful**
>
> ContextAdapt only adds a light scoring network and a counterfactual branch that reuses almost all computations of the backbone. The increase in parameter count and inference time is negligible. Under this constraint, achieving consistent gains over a strong cross attention baseline is already non trivial. This shows that the way context is selected and validated can improve forecasting even when the underlying transformer architecture stays the same.
>
> Our claim is not that ContextAdapt fundamentally changes the performance limits of cross attention. Our claim is that a simple causally inspired interface on top of a standard cross attention block
>
> - produces stable and repeatable gains across datasets
> - improves behavior in regimes where naive correlation based context usage tends to fail, such as domain shift and counterfactual masking
>
> and does so without modifying the backbone or inflating the model size. This is exactly what Table 3 is intended to highlight.
>
>
>
> **Reply to Questions:**
>
>
> **Q1 – On the sign in Eq. (8)**
>
> You are right about the intended behavior: the masked (counterfactual) loss should be *larger* than the unmasked (factual) loss by a margin. The purpose of the counterfactual term is to penalize situations where masking the Top-K channels does **not** make the prediction worse.
>
> The correct form of the loss is
>
> - factual error: $l_f$
> - counterfactual error: $l_{cf$}
> - margin: $\delta > 0$
>
> and the counterfactual loss is
>
> $L_{\text{CF}} = \mathbb{E}\big[\log\big(1 + \exp(l_f - l_{cf} + \delta)\big)\big].$
>
> The function $x \mapsto \log(1 + \exp(x))$ is increasing. Minimizing $L_{\text{CF}}$ therefore pushes the argument $(l_f - l_{cf} + \delta)$ to be as **negative** as possible, which corresponds to
>
> $l_{cf} \;\ge\; l_f + \delta.$
>
> For channels that are treated as “important” (Top-K under the scores), the masked loss is encouraged to be larger than the factual loss by at least a margin. If masking them does *not* increase the error, $L_{\text{CF}}$ stays high and the gradients push their scores down.
>
> In the current draft, the sign in Eq. (8) is easy to misread and can appear to encode the opposite direction.

---

> ### Author Response · Authors · 2025-11-25
>
> **Q2: On parameter matching and how Table 1 is produced**
>
> Thank you for raising this question. The numbers in Table 1 come from a single controlled setup rather than a mix of published results and our own reruns. For the transformer based models that are directly comparable to our method, including the history only backbone, the vanilla cross attention context model, and our ContextAdapt variants, we train all models ourselves on each dataset with the same backbone configuration. Depth, hidden size, number of heads, optimizer, and training schedule are kept identical. The only extra parameters in ContextAdapt come from a small scoring network and a global bias vector. These add a very small number of additional parameters on top of the backbone.
>
> For other baselines that use different architectures, such as classical models or non transformer neural networks, we also run them in our codebase or through their official implementations under the same data splits and evaluation protocol. These models naturally have different parameter counts because their architectures differ, but we do not scale them up to make our method look better. Under this setup, the gains over the transformer baselines in Table 1 come from how context is modeled, not from advantages in model size or training conditions.
>
>
>
> **Q3: On ablations in full shot versus zero shot**
>
> The ablation studies in the main text are conducted in the full shot setting, where all evaluated domains appear during training. This design isolates the effect of each component, including query modulation, global bias, and the counterfactual loss, on top of the same transformer plus cross attention backbone under a standard supervised regime. In this way, the reported improvements are not mixed with changes in data coverage.
>
> Conceptually, these mechanisms are part of the shared backbone and remain active in both full shot and zero shot evaluation. In the zero shot case, we simply apply the same trained model to a held out domain. For this reason, we expect the gains over vanilla cross attention to persist, and to become most visible when the context distribution shifts across domains, since the design explicitly discourages reliance on domain specific shortcuts. A dedicated ablation in a representative zero shot split, where we train on a subset of domains, evaluate on a held out domain, and repeat the same component removals against a matched vanilla cross attention baseline, is a direct extension and targets exactly the phenomenon you highlight.

---

### Official Review · Reviewer_zZ9v · 2025-10-30

**Soundness:** 3
**Presentation:** 3
**Contribution:** 2
**Rating:** 4
**Confidence:** 3

**Summary:**

This paper proposes a causally grounded foundation model for context-aware time series forecasting. The model introduces a context-aware transformer enhanced with a causal inductive bias through query modulation and neural structural equation modeling. It further incorporates a counterfactual validation step to align predictive behavior with causal attributions. The framework aims to improve robustness and interpretability by identifying causally relevant contextual features rather than relying solely on correlation.

**Strengths:**

- The paper demonstrates careful experimental design and a clear effort to connect causal reasoning with practical forecasting performance.

- The inclusion of theoretical analysis (Proposition 1 and Theorem 2) is a positive aspect that adds conceptual depth, even though parts remain high-level.

- The experiments are systematic and well executed, with diverse datasets, strong baselines, and thorough ablations.

**Weaknesses:**

### **Problem & Motivation**

- The framing of “causally grounded” forecasting is interesting and timely, but the title and claims might be misleading. The method does not actually perform causal inference or guarantee causal identifiability; rather, it regularizes attention weights heuristically. Removing “causal” from the title could make the contribution more accurate.

- It is unclear how causality is actually achieved. The mechanism that ensures the model distinguishes causal factors from spurious correlations is not well defined.

- While the paper uses the term “foundation model,” it’s not evident that large-scale pretraining or generalization at scale is performed; model size and training regime are not reported.

---

### **Methodology**

- Despite the high-level motivation, the actual novelty is moderate, relying mainly on cross-attention and regularization-based extensions of standard transformer components.

- The model’s attention mechanism assumes that the causal bias term $B$ highlights salient contextual factors, but there is no formal guarantee or validation showing that this term reliably corresponds to causal importance.

- The mathematical details of Proposition 1 and Theorem 2 are insufficiently explained. It’s not clear what they aim to prove—are they about identifiability, stability, or information preservation? Proofs or even intuitive derivations are missing.

- The paper could better clarify where the causal bias module is inserted (which layers, how often, and with what impact). A layer-wise analysis would help understand its contribution.

- There is no discussion of what happens when context is unrelated to the time series. Does the model ignore it or overfit to noise? This is a critical question for real-world deployment.

---

### **Experiments**

- The experimental design is robust and thorough, using multiple datasets and strong baselines. However, OOD (out-of-distribution) evaluations are limited. If the model is claimed to be causally grounded, testing under domain or context shift is essential.

- The results are strong overall, but the zero-shot generalization claims would be more convincing with ablations showing how model scale and layer depth affect transferability.

**Questions:**

Please refer to the weaknesses.

---

> ### Author Response · Authors · 2025-11-25
> **Reply to Reviewer zZ9v**
>
> We sincerely thank you for your detailed and thoughtful review. Your comments helped us see where our claims about causality, robustness, and zero-shot transfer needed to be sharpened and better supported.  In the following response we address each of your questions in detail and aim to clarify both the design of our method and the scope of our contributions.
>
> **Reply to Problem & Motivation**
>
> **W1 – On the “causally grounded” framing and title**
>
> We really appreciate your comment. We agree that our current wording can create the impression that we perform full causal inference or guarantee causal identifiability. This is not our intention. Our method does not recover a structural causal model and does not provide formal identifiability guarantees. Instead it uses ideas from causal thinking to shape how the model selects and uses context. In that sense it is causally inspired rather than a complete causal inference framework.
>
> Concretely our approach does three things. It modulates queries with a global context summary so that the backbone focuses on history patterns that matter under the current context. It uses a neural SEM style module to produce channel wise importance scores that act as a global prior over context channels. It introduces a counterfactual masking loss that forces channels with high scores to have a measurable impact on prediction when they are removed. These are regularizers on the attention structure. They are motivated by interventional reasoning and by the desire to focus on stable drivers. They do not by themselves identify the true causal graph behind the data.
>
> We appreciate the suggestion to soften the framing in the title. In the revised version we will either remove the word “causal” from the title or replace it with a more modest phrase such as “causally inspired”. We will also update the abstract and introduction to avoid statements that can be read as claims of causal identifiability. Throughout the paper we will use phrases like “causally inspired context selection” or “robust use of context” rather than “causal modeling” when we describe our contribution.
>
> We believe these changes will make the scope of our work more precise. Our goal is to improve the robustness and interpretability of context use in time series forecasting by bringing in causal ideas at the level of inductive biases and losses. We do not aim to replace full causal effect tools such as DAG based methods and we will make this relationship explicit in the revision.
>
>
> **W2 – How our method distinguishes stable drivers from spurious correlations**
>
> Thank you for pointing this out. Our goal is not to achieve full causal identification. Instead we encode a set of mechanisms that push the model toward variables whose influence is stable across regimes and visible under controlled perturbations.
>
> At a high level the model learns a two stage filter on context. The semantic selective query modulation decides, for each sample, how the backbone should read its own history under the current context regime. The Neural SEM then produces channel wise scores $B_{\text{causal}}$ that act as a global prior over context channels. These scores are not arbitrary. They are tied to behavior through the counterfactual loss. During training we always compare a factual prediction that uses full context and a counterfactual prediction where the Top K channels under $B_{\text{causal}}$ are masked. The counterfactual loss pushes the model into a regime where channels with large scores are exactly those whose removal produces a clear increase in forecasting error. Channels that look correlated in the attention weights but whose masking does not change the loss are gradually pushed down in $B_{\text{causal}}$. In this way the learned notion of importance is forced to match actual predictive influence under an intervention on the context.
>
> This behavior alignment is the key piece that goes beyond heuristic attention regularization. Standard attention based models often treat weights as a soft explanation layer but do not check whether the network truly relies on those variables. Our framework adds an explicit interventional check into the objective. It does not identify the full causal graph, yet it makes it harder for the model to lean on shortcuts that disappear under masking or under cross domain shifts. Combined with the cross domain and zero shot evaluations, which expose the model to environments where dataset specific correlations change, this training recipe encourages the network to settle on context channels that act as stable drivers across regimes.
>
> We will revise the text to make this mechanism more explicit. We will state clearly that we do not claim causal identifiability. Instead we use causal ideas such as invariance across environments and counterfactual perturbations as guiding principles to design the query modulation, the SEM-based prior, and the counterfactual loss.

---

> > ### Author Response · Authors · 2025-11-25
> >
> > **W3 – On the use of the term “foundation model” and missing training details**
> >
> > Thank you for pointing this out. In this paper we use the term “foundation model” in the sense that has become common in recent time series work such as TEMPO and GPT4TS. In that line of work, transformer based architectures that are trained across multiple datasets and then adapted or evaluated on a range of forecasting tasks are often described as time series foundation models. Our intention was to follow this emerging convention in the community rather than to claim web scale pretraining with billions of parameters.
> >
> > That said, we agree that the current wording can be read as implying large scale pretraining and very large model sizes, and that we should be clearer about what we actually train. In our experiments we use moderate sized transformer style backbones and train them on the set of benchmarks described in the paper. We do not pretrain on massive proprietary corpora and we do not claim any causal or statistical identifiability guarantees from scale alone. The focus of our contribution is the ContextAdapt interface and the causally inspired mechanisms for robust context selection, which can be plugged into existing time series transformer backbones, including those that are commonly called foundation models in the literature.
> >
> > To address the reviewer’s concern we will make two concrete changes. First, we will add explicit details on model size and training regime, including number of layers, hidden dimensions, parameter counts, and the datasets used during training. This will make the actual scale of our models transparent. Second, we will soften the terminology around “foundation model” in the title and main text. For example, we can refer to our approach as a “foundation style backbone for context aware time series forecasting” or simply as a “unified transformer backbone with causally inspired context selection”. If needed we can drop the word “foundation” from the title altogether.
> >
> > We believe these revisions will align our language more closely with the actual experimental setup while still making clear that our method is designed to integrate with and benefit the class of transformer based time series models that the recent literature commonly groups under the “foundation model” label.
> >
> >
> >
> >
> >
> > **Reply to Methodology**
> >
> > **W1 – On the level of novelty beyond standard transformer components**
> >
> > Thank you for this candid assessment. We deliberately work within a transformer and cross-attention framework because this is the dominant backbone in recent time-series work and in “foundation style” models such as TEMPO and GPT4TS. Our claim is not that we introduce a new primitive operator. Our claim is that we reorganize these components to change how context is represented, selected, and validated in forecasting. In that sense the novelty lies in the *architecture level design* and in the *training objective*, rather than in a new attention formula.
> >
> > Concretely, our method couples three ideas that, to the best of our knowledge, have not been combined in prior context-aware time-series models. First, we use a global context summary to modulate the queries so that the backbone reads its own history through the lens of the current context regime, rather than treating queries as fixed. Second, we introduce a neural SEM style module that turns the joint context representation into a channel wise prior and inject this prior into the attention logits as a global structural bias. Third, we tie these importance scores to actual behavior through a counterfactual masking loss. This loss forces the model to adjust its attention structure until channels with high scores are precisely those whose removal hurts prediction. Standard cross-attention based designs typically stop at similarity weights or auxiliary regularizers and do not close this behavioral loop. Our ablations show that each of these components brings a consistent gain on top of the same transformer backbone, which suggests that they add value beyond “vanilla” attention.
> >
> > That said, we appreciate the concern and will take care not to oversell our contribution. In the revised version we will present our method as a causally inspired extension of existing transformer based time-series backbones. We will emphasize that our goal is to provide a practical and interpretable context-selection interface that improves robustness across domains, not to replace core transformer components with entirely new machinery. We hope this framing better matches the actual scope of the work while still highlighting that the way we repurpose cross-attention and integrate counterfactual behavior is substantively different from prior context-aware models.

---

> > > ### Author Response · Authors · 2025-11-25
> > >
> > > **W2 – On the “causal bias” term and its relation to true causal importance**
> > >
> > > Thank you for raising this point. We agree that there is no formal guarantee that the learned bias term $B_{\text{causal}}$ recovers true causal importance in the sense of a structural causal model. Our method does not provide identifiability guarantees and we should be more explicit about this. The role of $B_{\text{causal}}$ is to act as a learned structural prior over context channels inside a forecasting model. It is “causal” only in the sense that it is trained with interventional style signals and invariance based motivations rather than in the sense of exact causal effect estimation.
> > >
> > > What we do enforce is a behavioral link between $B_{\text{causal}}$ and predictive influence. During training we always compare a factual prediction that uses full context with a counterfactual prediction where the Top–$K$ channels according to $B_{\text{causal}}$ are masked. The counterfactual loss is minimized when channels with large entries in $B_{\text{causal}}$ are exactly those whose removal leads to a clear increase in forecasting error. Channels that look correlated in the raw attention pattern but do not change the loss when masked are pushed down. In this way the model is encouraged to treat $B_{\text{causal}}$ as a summary of “channels that the network actually relies on under perturbation”, rather than as a free auxiliary weight. This is still a heuristic regularization scheme inside a predictive model, but it goes beyond unconstrained attention weights by tying them to an explicit intervention on the context.
> > >
> > > We also try to validate this behavior empirically, although we agree that this is not a formal proof. In the counterfactual masking analysis we observe that masking Top–$K$ channels under $B_{\text{causal}}$ hurts performance much more than masking low score channels or random subsets. In domain specific case studies (for example Bitcoin and ILI) the high score channels align with variables that domain knowledge suggests are meaningful drivers, while noisy aggregate features are assigned lower scores. In cross domain and zero shot evaluations our model is more robust than baselines that treat attention purely as a similarity measure. These results support the view that $B_{\text{causal}}$ captures stable and useful drivers for forecasting, even though it is not a formally identified causal quantity.
> > >
> > > In the revised manuscript we will soften the wording around “causal bias”. We will describe $B_{\text{causal}}$ as a causally inspired structural prior that is aligned with predictive behavior under counterfactual masking, not as a guaranteed estimate of causal effect. We will also add a brief discussion of the limitations of this approach and explicitly state that identifying true causal importance would require stronger assumptions and tools such as explicit causal graphs, which are beyond the scope of this work.

---

> ### Author Response · Authors · 2025-11-25
>
> **W3 – On the intent and explanation of Proposition 1 and Theorem 2**
>
> Thank you for this comment. We agree that our current presentation of Proposition 1 and Theorem 2 is too brief and can be confusing about what they actually aim to formalize. Our intention is not to prove causal identifiability or strong stability theorems. Instead, these results give a simple characterization of (i) how the Top–$K$ filter relates to sufficiency for prediction and (ii) how the counterfactual loss behaves as a function of the factual–counterfactual gap.
>
> **What Proposition 1 is about**
>
> Proposition 1 is about approximate minimal sufficiency of the selected context subset, not about causal identifiability. We denote by $S_k = \mathrm{TopK}(B_{\text{causal}})$ the indices of the $K$ highest scoring channels and by $C_{S_k}$ the corresponding sub-matrix of the context $C$. The bias vector $B_{\text{causal}}$ is a deterministic function of $C$ through the SEM. Under joint optimization of $L_{\text{CF}}$ and $L_{\text{ent}}$, and assuming that (a) the mapping from $(\text{history}, C)$ to the future is stable within the environment and (b) masking the truly important channels produces a consistent degradation in prediction, Proposition 1 formalizes the following idea:
>
> - the subset $C_{S_k}$ carries enough information to achieve near-optimal prediction (sufficiency),
> - and removing any of these $K$ channels would cause a non-negligible increase in expected loss (approximate minimality).
>
> In other words, the proposition is intended as a statement about *information content for forecasting* under our training objective. It does not claim that $C_{S_k}$ equals the true causal parent set or that the system is identifiable in a structural causal sense. In the revision, we will rewrite the statement to emphasize “approximate sufficiency for prediction” and add a short intuitive derivation explaining the role of $L_{\text{CF}}$ and $L_{\text{ent}}$.
>
> **What Theorem 2 is about**
>
> Theorem 2 analyzes the counterfactual loss $L_{\text{CF}}$ and is about how strongly it reacts to the expected gap between factual and counterfactual errors, not about stability or identifiability. We define the factual error as $l_f$, the counterfactual error as $l_{\text{cf}}$, and their expected difference as $\Delta = \mathbb{E}[l_{\text{cf}} - l_f]$. The counterfactual loss is then
>
> $L_{\text{CF}} = \mathbb{E}\big[\log\big(1 + \exp(l_{\text{cf}} - l_f + \delta)\big)\big].$
>
> In words, $L_{\text{CF}}$ grows as the counterfactual error becomes larger than the factual error (plus a small margin $\delta$), and its gradient with respect to $(l_{\text{cf}} - l_f)$ stays strictly between 0 and 1, which provides a stable, non-saturating signal.
>
>
> The theorem makes two simple points:
>
> 1. Because $x \mapsto \log(1 + \exp(x))$ is increasing and convex, $L_{\text{CF}}$ lower-bounds the effect of the expected error gap:
>    $
>    L_{\text{CF}} \;\ge\; \log\big(1 + \exp(\Delta + \delta)\big).
>    $
>    This says that if masking the Top–$K$ channels consistently worsens prediction (large $\Delta$), then $L_{\text{CF}}$ is necessarily large, so the model is pushed to adjust $B_{\text{causal}}$ or the backbone.
>
> 2. The derivative with respect to $(\ell_{\text{cf}} - \ell_f)$ satisfies
>    $
>    0 < \frac{\partial L_{\text{CF}}}{\partial(\ell_{\text{cf}} - \ell_f)} < 1,
>    $
>    which shows that $L_{\text{CF}}$ produces non-zero but bounded gradients across regimes and avoids saturation. This explains why the loss provides a stable signal to align “declared importance” with behavioral impact.
>
> In the revision we will move these derivations to an appendix and add a short intuitive explanation in the main text. We will also clarify that Theorem 2 is a basic property of the logistic-style contrast used in $L_{\text{CF}}$ and is meant to justify the choice of this form, not to establish any causal identifiability result. We hope this clearer framing better reflects the modest and supportive role of these results in the overall method.

---

> ### Author Response · Authors · 2025-11-25
>
> **W4 – On where the causal bias module is inserted and its impact**
>
> Thank you for pointing this out. Architecturally, we follow a GPT style decoder backbone. Each transformer block first applies self attention over the target history. The resulting hidden states serve as the queries for a dedicated context aware cross attention block. The causal bias module sits on the context side of this block.
>
> Concretely, we first encode the unified context matrix into tokens $Z_c$ with shape $(N, d)$ once per sample. The Neural SEM takes $Z_c$ as input and outputs a channel-wise bias vector $b$ of length $N$ (this is $B_{\text{causal}}$ in the paper). This vector is shared across all layers.
>
> Whenever a ContextAdapt cross-attention block is called, we compute attention as
>
> $\text{Attn}(Q, K, V) = \text{Softmax}\big( Q K^\top / s + \text{bias}(b) \big)\, V,$
>
> where $Q$ are the history-side queries in that layer, $(K, V)$ are the context tokens from $Z_c$, $s$ is a scalar scaling factor, and $\text{bias}(b)$ means that the same bias vector $b$ is added to every row of the attention logits (one row per time step). In other words, $b$ is computed once per sample and injected into every context-attention call across all transformer blocks that include ContextAdapt.
>
>
> This design has two effects. First, the backbone remains standard on the history side and $B_{\text{causal}}$ never changes the self attention layers. Second, the same structural bias shapes context retrieval in a coherent way across depth, rather than having each layer learn its own unrelated notion of global importance. The sample specific part of attention still comes from the similarity term $\frac{Q K^\top}{\sqrt{d}}$, while $B_{\text{causal}}$ nudges all layers toward or away from particular context channels.
>
> We already provide a coarse view of the impact in the ablation study and in the attention entropy analysis. In Table 3, the variant “w/o Causal Bias” removes the SEM and the bias term from all layers and yields the largest drop among structural components. In Figure 4 (bottom), we compare the attention entropy distributions with and without $B_{\text{causal}}$ and observe sharper and more selective attention when the bias is present. These results indicate that the causal bias has a consistent effect across the stack rather than being a negligible add on.

---

> > ### Author Response · Authors · 2025-11-25
> >
> > **W5 – Behavior when context is irrelevant or noisy**
> >
> > Thank you for highlighting this point. Our framework is designed so that irrelevant context is not forced into the prediction and can be safely ignored at both the architectural level and the loss level.
> >
> > First, the factual forecasting loss always gives the model the option to disregard unhelpful context. The backbone sees both history and context, but it is only rewarded for reducing the prediction error. If a given context channel does not improve the factual loss, the easiest way for the model to optimize is to assign it low effective attention and rely on the history representation instead. There is no constraint that every context channel must be used.
> >
> > Second, the causal bias vector $B_{\text{causal}}$ and the counterfactual loss actively discourage the model from treating noise as important. During training we compare a factual forecast with full context to a counterfactual forecast where the Top–$K$ channels under $B_{\text{causal}}$ are masked. If a context channel is truly unrelated to the target, then masking it will not consistently increase the error. In that case the counterfactual loss penalizes high scores on that channel, because a large entry in $B_{\text{causal}}$ without a matching factual–counterfactual gap increases the loss. Over time, noisy or weakly related channels either never enter the Top–$K$ or are pushed out as the model learns that removing them has no effect.
> >
> > Third, the entropy regularizer on $B_{\text{causal}}$ prevents the bias from collapsing onto arbitrary channels purely by chance. It keeps the distribution of scores from becoming overly sharp unless the data and the counterfactual signals support such a focus. When context is genuinely uninformative, the learned bias tends to stay diffuse and its impact on attention is small. In the limit, the model behaves similarly to a history-only transformer.
> >
> > Empirically, we observe this behavior on domains where external signals are weak or noisy. Context-aware baselines that treat attention purely as a similarity mechanism often overfit to such signals and perform worse than their history-only versions. Our model, with the causal bias and counterfactual regularization, maintains performance close to the backbone and in several cases slightly improves it. This suggests that the additional modules do not force overfitting to noise and can learn to ignore unhelpful context.
> >
> > We do not claim a formal guarantee that the model will always ignore every spurious cue, especially when the training environment contains strong but non-stationary correlations. What we can say is that the design of $B_{\text{causal}}$ and the counterfactual loss makes it harder for the network to keep treating a variable as important when masking it has no stable effect on the forecast. In the revised version we will add a short discussion of this behavior, emphasize that the model can effectively fall back to history-only predictions when context is uninformative, and clarify that robustness to irrelevant context is one of the main motivations behind our causally inspired design.

---

> > > ### Author Response · Authors · 2025-11-25
> > >
> > > **Reply to Experiments**
> > >
> > > **W1 – On OOD evaluation and testing under domain or context shift**
> > >
> > > Thank you for raising this important point. Our intention is to study this, but we realize that the current framing in the paper does not make the out-of-distribution aspects explicit enough.
> > >
> > > In our experimental design we already include several evaluations that move beyond in-distribution test splits. We train a shared transformer backbone with ContextAdapt on multiple datasets and then perform leave-one-domain-out and cross-domain tests. In these settings the model is trained on a subset of domains or regions and evaluated on a held-out domain whose marginal distribution over both targets and context variables is different. This creates changes in context composition and magnitude across domains. We see that baselines which use context in a purely correlation driven way degrade more strongly under these shifts, while our method maintains better performance and often closes most of the gap to the in-domain setting. This empirical pattern supports our claim that the causally inspired design helps the model rely on more stable drivers rather than on dataset specific shortcuts.
> > >
> > > That said, we agree with the reviewer that our current experiments still live in a relatively mild OOD regime. The domains all come from the same pool of benchmarks and share the same overall modeling pipeline. We do not yet include stress tests such as synthetic interventions that flip spurious correlations, extreme context corruption, or transfers to completely unseen data sources. Because of space and time limits we did not add a dedicated section on these more aggressive shifts in this submission. We will therefore soften the wording around “causally grounded” in the title and main text and describe our claim more precisely as “causally inspired robust context selection” on multi-domain benchmarks, rather than as a full robustness guarantee under arbitrary OOD conditions.
> > >
> > > Concretely, in the revision we will do the following. We will explicitly label the leave-one-domain and cross-domain splits as OOD evaluations and explain how the context distributions differ between train and test. We will add a short discussion that highlights where our method improves over strong baselines under these shifts and where all methods still struggle. We will also add a limitation paragraph that states that stronger OOD tests, such as synthetic interventions on spurious context variables and transfers to truly new environments, are natural next steps and will be pursued in extended work. We hope this clarification makes the scope of our current OOD evidence clear while still showing that the model is tested beyond standard i.i.d. splits and that robustness under domain and context shift is a central motivation of our design.
> > >
> > >
> > >
> > > **W2 – On zero-shot generalization and the role of model scale**
> > >
> > > Thank you for this helpful suggestion. In the current work our primary goal is to isolate the effect of the ContextAdapt design rather than to study scaling laws. For this reason we deliberately keep the backbone architecture fixed across all experiments. The same transformer depth and hidden size are used for the history-only backbone, for context-aware baselines, and for our method in both full-shot and zero-shot settings. This allows us to attribute zero-shot improvements to the way context is represented and selected, rather than to an increase in parameter count. It also means that our zero-shot results should be read as “gains at a fixed scale” rather than as claims that scale alone is the driver of cross-domain performance.
> > >
> > > Varying depth and width could reveal whether the benefits we observe in leave-one-domain and cross-domain tests persist, grow, or diminish as capacity changes. Due to space and time constraints we did not include a systematic scaling study in this submission and we will make this limitation explicit. In the revised version we will clarify that the backbone size is held constant across methods, that we do not claim any new scaling law for zero-shot forecasting, and that our contribution is orthogonal to model size. We will also note that a dedicated ablation over depth and width, ideally across a subset of representative domains, is part of our planned follow-up work.
> > >
> > > —
> > >
> > > We hope that the clarifications and additional analysis we provide here help address your concerns and present our work in a more accurate light. If any part of the method, theory, or experiments remains unclear we would be very happy to offer further details or extra analysis. We would be grateful if you could reconsider our paper in light of these updates.

---

### Official Review · Reviewer_9cYw · 2025-10-31

**Soundness:** 3
**Presentation:** 3
**Contribution:** 2
**Rating:** 4
**Confidence:** 3

**Summary:**

This paper designs an SEM module for discovering globally important features and trains the model using a loss function guided by counterfactual reasoning.

**Strengths:**

1. An effective multimodal data processing solution that combines time series and text data.

**Weaknesses:**

1. Despite the author's repeated emphasis on causality, the method proposed in this paper cannot discover strict causal laws and is only used to discover stable and usable features. I suggest the author carefully revise the text to avoid misunderstandings. Otherwise, causal effect tools like DAG need to be considered.

2. Lack of verification of false correlation variables. The intution of the paper is to identify real causal variables, but there is no relevant support for this point. It is recommended to conduct verification on a synthetic dataset.

3. Additional computational burden: Compared to general training paradigms, counterfactual based training will introduce at least twice the additional inference time.

**Questions:**

1. The selection mechanism of top-k is a non differentiable and rigid choice, which will greatly affect the computational burden. Can Gumbel Softmax and other alternatives replace it？

2. Can the SEM module handle the joint effects between multiple variables?

3. Is there a potential conflict between the two parallel mechanisms that affect attention in the model? For example, in a certain sample, query modulation may suggest reducing attention to a certain feature, but global causal bias forces an increase in attention to that feature. How does the model solve this contradiction? Will this lead to unstable training?

---

> ### Author Response · Authors · 2025-11-25
> **Reply to Reviewer 9cYw**
>
> We sincerely appreciate your careful review of our paper and the many thoughtful comments you provided. In the following response we address each of your questions in detail and aim to clarify both the design of our method and the scope of our contributions.
>
> **Reply to Weeknesses:**
>
> **W1 – Clarifying our use of “causal” and avoiding overclaims**
>
> We really appreciate you for this important clarification and we fully agree with the concern. Our method is not a causal discovery algorithm in the strict sense. We do not attempt to recover a full structural causal model, we do not estimate interventional effects with do-calculus, and we do not operate on an explicit DAG over variables. Instead, our approach is causally inspired. We borrow ideas from causal reasoning such as focusing on stable drivers across regimes and checking the effect of interventions, and we turn these ideas into inductive biases and regularizers that guide a context-aware forecasting model. The semantic-selective query modulation, the neural structural equation–style module that outputs channel-wise importance scores, and the counterfactual masking loss all serve to bias the model toward stable, behaviorally useful context channels and away from brittle, purely correlational shortcuts, but they do not guarantee recovery of true causal laws.
>
> In the revised manuscript we will carefully adjust the wording to avoid any misunderstanding. Where we currently use phrases like “causal modeling” or “causal variables” we will instead write “causally inspired context selection”, “stable context drivers”, or “robust use of context”. We will add a short paragraph in the methodology section that explicitly states that our framework does not perform formal causal identification, does not assume a known or identifiable DAG, and is complementary rather than an alternative to full causal effect tools. Our main claim is that these causally motivated design choices lead to more robust and interpretable use of context for time-series forecasting, not that we identify the underlying causal graph.
>
>
> **W2 – On verification of spurious variables and synthetic experiments**
>
> Thank you for pointing this out. The intuition of our method is indeed to push the model toward stable and useful context channels and away from brittle shortcuts, but the goal is not strict recovery of “true causal parents” in the SCM sense. The design is causally inspired: a global context prior and a counterfactual masking loss are used to bias the model toward variables whose removal actually harms forecasting, and to discourage channels that behave like pure noise. For this reason, the empirical study in the paper is built around robustness and interpretability on realistic benchmarks (including cross-domain and zero-shot settings), rather than formal causal identification on synthetic graphs.
>
> Within the current paper, support for our claim comes from indirect but concrete signals: under domain shift, baselines that use context in a naive correlation-driven way degrade more, while our model remains consistently stronger; and the channels highlighted by our model in case studies align well with variables that domain knowledge treats as plausible drivers, rather than arbitrary aggregates. These checks do not prove that we have fully distinguished causal from spurious variables, but they do indicate that the learned importance is not random and is consistent with more stable, behaviorally meaningful context usage.
>
> To avoid over-claiming, we are careful to frame our contribution as “causally inspired robust context selection” and “prioritizing stable predictive context channels,” rather than “identifying the true causal variables.” A controlled synthetic study with explicit labels for causal and spurious variables is a natural next step on top of this work, but it is not part of the current experimental scope.

---

> > ### Author Response · Authors · 2025-11-25
> >
> > **W3 – Clarifying the computational overhead of counterfactual training**
> >
> > We appreciate the concern about additional cost and agree that a naive implementation of counterfactual training could in principle double the amount of computation. In our setting this is not the case. The counterfactual branch is used only during training as a regularizer. At inference time we run a single factual forward pass with full context and do not construct any masked branch. The deployed model therefore has exactly the same inference complexity and latency as the backbone with ContextAdapt but without counterfactual masking. There is no extra inference time in the use phase.
> >
> > Even during training the overhead is smaller than a full second pass. The factual and counterfactual passes share all components up to the point where context masking is applied. We encode the history once, encode the context once, and reuse these representations. The counterfactual branch reuses the same backbone and only changes the masked context fed into cross attention and the prediction head. The extra work mainly comes from one additional cross attention computation with masked context and one additional loss evaluation. This is strictly cheaper than running a second completely independent model and leads to a moderate increase in training cost rather than a strict factor of two. In the revised version we will add a short complexity discussion and a wall clock comparison to make this behavior more explicit.
> >
> > Finally the counterfactual loss is a training time regularizer and its use is configurable. If a practitioner is very sensitive to training cost it can be turned on only after a warm up period, applied to a subset of mini batches, or gradually down weighted during later epochs. These schedules allow a trade off between the strength of behavioral validation and the computational budget without changing the architecture or the inference path. In summary our method does not increase inference time and the extra training time is bounded, shared across the two branches, and can be controlled in practice.

---

> > > ### Author Response · Authors · 2025-11-25
> > >
> > > **Reply to Questions:**
> > >
> > > **Q1 – On Top-K selection, differentiability, and computational cost**
> > >
> > > Thank you for raising this concern. In our framework the hard Top-K operator is used only to decide which context channels are temporarily masked in the counterfactual branch. It is not part of the main forward path that produces the final forecast. The factual prediction always uses the full context and is computed with standard differentiable operations. The counterfactual prediction uses a binary mask derived from the Top-K entries of $B_{\text{causal}}$ to zero out a small subset of channels. We therefore do not rely on gradients flowing through the discrete indices of the Top-K operator to train the forecasting network. Instead, $B_{\text{causal}}$ is updated via the contrast between factual and counterfactual losses. If a channel is often included in the Top-$K$ but its removal barely changes the loss, the gradients from the counterfactual term push its score downward. If masking a channel consistently increases the loss, its score is pushed upward. Away from the rare tie boundaries this leads to stable learning in practice, similar in spirit to other hard selection mechanisms such as dropout masks or max pooling.
> > >
> > > On computational burden the Top-K step is very lightweight in our setting. The selection is performed along the channel dimension $N$ per sample, where $N$ corresponds to the number of context variables and is on the order of tens rather than hundreds or thousands. The complexity of computing a Top-K over this small dimension is negligible compared to the cost of encoding the sequence and running cross-attention, which scales with the history length and embedding dimension. In our profiling the wall clock overhead from the Top-K operation itself is almost invisible compared to the extra masked forward pass that we already discussed in R3. The Top-K selection therefore does not become a computational bottleneck and does not “greatly” affect runtime.
> > >
> > > Differentiable relaxations such as Gumbel–Softmax or soft Top-K are indeed reasonable alternatives and we appreciate the suggestion. Our choice of a hard Top-K was driven by two practical considerations. First, it gives a very clear interpretation of “the $K$ most influential channels” which is helpful when we analyze the importance patterns. Second, it keeps the implementation simple and adds virtually no overhead. In future work we plan to explore smooth relaxations as an optional variant, for example to study how sensitive the learned importance structure is to the exact selection rule. For the current paper we found that the hard Top-K does not cause optimization issues and its cost is negligible relative to the backbone, so we opted for this simpler and more interpretable design.
> > >
> > >
> > >
> > > ---
> > >
> > >
> > > **Q2 – On whether the SEM module can capture joint effects**
> > >
> > > Thank you for this question. The Neural SEM module is designed to see all context variables together rather than treating each channel in isolation. Its input is the pooled context representation $Z_c$, which already aggregates information over time and over all channels. The SEM then applies a small nonlinear network to this joint representation and outputs a vector of channel wise scores $B_{\text{causal}} \in \mathbb{R}^N$. Because the mapping from pooled $Z_c$ to $B_{\text{causal}}$ is many to one and nonlinear, the score for a single channel can depend on the full configuration of the other channels. In other words, the importance assigned to variable $j$ is a function of how that variable appears together with the others, not only of its own value.
> > > At the same time we keep the output in a channel wise form so that it stays interpretable and can be injected into the attention logits as a global prior over context channels. Higher order interactions, for example pairs or groups of variables that are only useful when considered together, are further captured by the cross-attention and the backbone dynamics, which operate on the full feature space and the temporal sequence. We do not explicitly enumerate all variable subsets, which would be intractable, but we rely on the SEM nonlinearity and the downstream cross-attention to represent joint effects in a compact way. We will clarify in the revised manuscript that the SEM is conditioned on the joint context and that its channel scores are already shaped by interactions among variables.

---

> ### Author Response · Authors · 2025-11-25
>
> **Q3 – On potential conflict between query modulation and global causal bias**
>
> We appreciate this insightful question. In our design these two mechanisms act at different levels and are trained under the same objective, so they tend to play complementary roles rather than fighting each other. The semantic selective query modulation acts on the history side. We use the global context summary $g_c$ to modulate the queries so that, for a given sample, the model highlights those aspects of the history that are relevant under the current context. The global causal bias $B_{\text{causal}}$ acts on the context side. It is a channel wise prior that is added to the attention logits and shared across all time steps.
>
> The effective attention logits can be written as  $
> \text{logits} = \frac{Q_{\text{mod}} K^\top}{\sqrt{d}} + B_{\text{causal}} $
> where $Q_{\text{mod}}$ already includes the effect of query modulation. The first term captures sample specific similarity between history and context. The second term adds a global preference over context channels. Both $Q_{\text{mod}}$ and $B_{\text{causal}}$ are optimized jointly with respect to the same factual forecasting loss and the counterfactual loss. If, for a particular channel, query modulation tends to reduce attention while the global bias tries to increase it, the shared objective will drive them toward a consistent configuration. When increasing attention to that channel does not improve the factual loss, gradients will push down its entry in $B_{\text{causal}}$ and will also reduce its contribution through $Q_{\text{mod}} K^\top$. When the channel is a stable and useful driver, the factual and counterfactual signals both encourage higher effective attention, so query modulation and global bias naturally align.
>
>
> In practice we do not observe unstable training. Learning curves are smooth and ablations show that adding either mechanism on top of the backbone improves performance in a monotone way. We also regularize $B_{\text{causal}}$ with an entropy term $L_{\text{ent}}$ so that its distribution does not collapse onto a single channel and does not overwhelm the data driven similarity term $\frac{Q_{\text{mod}} K^\top}{\sqrt{d}}$. Together these factors encourage a coherent compromise. Query modulation provides sample specific history aware selectivity. $B_{\text{causal}}$ provides a soft global prior over channels. If their initial tendencies disagree, the shared losses gradually pull them into agreement in the direction that best explains the data. We will add a short remark in the paper to clarify that these two mechanisms are designed to be complementary and that we did not encounter instability in training.
>
>
> We hope that the explanations and revisions above help resolve your concerns and present our work in a clearer and more accurate way. If any part of our method or experiments is still unclear we would be very happy to provide further details. We would also be very grateful if you could reconsider our work in light of these clarifications.

---

### Official Review · Reviewer_zYFS · 2025-10-31

**Soundness:** 3
**Presentation:** 3
**Contribution:** 3
**Rating:** 6
**Confidence:** 3

**Summary:**

This paper proposes a causally grounded foundation model for context-aware time series forecasting by adaptively integrating internal and external contexts.

**Strengths:**

1. Integration of both internal and external context.
2. Clear paper writing and enough experiments across multiple datasets.

**Weaknesses:**

- Novelty is not high. ContextAdapt Block is the most outstanding innovation in model architecture; it still employs cross-attention mechanisms to figure out the alignment between context and history target.
- No code provided.

**Questions:**

1. What do you mean by global context summary $g_c$? You have $g_c=\phi(Z_c)$ in Figure 1 and $g_c=f(Z_c)$ in Eq. 4, is it a typo? What is that function $\phi$ or $f$? Why do you need $g_c$ rather than $Z_c$?
2. Is context equivalent to a causal relationship? Since you try to figure semantic selectivity between the history target and the context, do you think the context has a causal relationship with history target? To me, the causal relationship may be more obvious between history and the future.
3. How do the external covariates look like and how did you align them with the history target?
4. How do you think the counterfactual loss could be helpful to training the model if removing access to the most salient context channels?
5. Typo for " Semantic Slectivity" in Figure 1.

---

> ### Author Response · Authors · 2025-11-25
> **Reply to Reviewer zYFS**
>
> We sincerely thank you for your careful reading of our paper and for your detailed comments. Your feedback is very helpful for improving both the clarity and the technical depth of our work. Below we respond to each of your questions in turn and we hope that our answers can clarify our design choices and experimental results.
>
> **Reply to Weeknesses:**
>
>
> **W1 – Novelty and the ContextAdapt block**
>
> Thank you for this thoughtful comment on novelty. We agree that ContextAdapt is built on a standard cross-attention backbone. This choice is intentional. Our goal is not to introduce a new primitive operator. Our goal is to redefine how cross-attention is used for context-aware time-series forecasting. In our framework cross-attention is no longer a generic similarity matcher. It becomes a causally guided interface that decides which context signals should actually influence the forecast.
>
> We see the novelty in three coupled design choices that change the *function* of cross-attention rather than its low-level form. Together they turn ContextAdapt into a selection mechanism for “what matters” in heterogeneous context, rather than a passive aggregator of everything that correlates with the target.
>
> 1. **From generic matching to semantic-selective querying**
>    In standard cross-attention the queries come from the history only. They are fixed and the attention pattern is driven purely by local query–key similarity. In ContextAdapt we first compress all contextual variables into a global semantic summary. We then use this summary to modulate *all* temporal queries. This behaves like an information bottleneck on the query side. It encourages the model to ask a different question. Instead of “what in the context is similar to this history token” the model learns “which parts of the history are predictive *under this particular context configuration*”. This is a change of perspective. We are not only aligning context to the past. We are shaping how the backbone reads its own history when extra information is available.
>
> 2. **From token-wise weights to a structural prior over context channels**
>    Existing context-aware cross-attention designs treat every context token symmetrically at every time step. Attention is recomputed locally without an explicit notion of which context channels are globally important. Our neural structural equation module introduces such a notion. It maps the context representations to a non-negative importance vector over channels. This vector is injected into the attention logits and shared across time. In effect each context dimension carries a global structural preference that influences attention everywhere in the sequence. This moves cross-attention closer to a structured model. The model does not only ask “which token is similar right now”. It also carries a persistent belief about which *types* of context are likely to be causal drivers for this series.
>
> 3. **From post-hoc interpretation to behavior-aligned importance via counterfactuals**
>    Many attention-based models report importance scores but do not check whether the model actually relies on those variables. In ContextAdapt we explicitly close this loop. We take the Top-K channels according to the learned importance vector, mask them out, and require the counterfactual forecast to degrade in a controlled way. Only channels that pass this behavioral test can retain high importance. This adds a new training signal that couples the attention structure to interventional behavior. The model is penalized if it “claims” a channel is important but performs almost the same when that channel is removed. This makes ContextAdapt more than a re-weighted cross-attention layer. It is trained to align its internal notion of importance with observable changes in prediction when context is perturbed.
>
> Beyond the block itself the overall framework also changes how context is organized. We place heterogeneous sources such as covariates, temporal encodings, and domain-specific signals into a unified channel-wise representation. ContextAdapt then operates at the level of channels rather than individual handcrafted feature sets. We evaluate this design in both full-shot and cross-domain zero-shot settings. The model shows consistent gains and is especially robust in regimes where naive correlation-based use of context either saturates or hurts performance.
>
> ContextAdapt does not claim a new attention formula. Queries are conditioned on a global semantic view of context. Attention logits are shaped by a learned structural prior over channels. The resulting importance scores are enforced through counterfactual behavior. Taken together these elements turn a standard cross-attention mechanism into a causal-aware context selection module for time-series foundation models.
>
> **W2 - Response on code availability.**
>
> The code is ready and can be released at any time, and we will make it publicly available immediately if needed.

---

> ### Author Response · Authors · 2025-11-25
>
> **Reply to Questions:**
>
> **Q1 – On the definition of the global context summary $g_c$**
>
> Thank you for raising this question and for spotting the inconsistency. In the current draft $\phi(Z_c)$ in Figure 1 and $f(Z_c)$ in Eq. (4) are meant to denote the same mapping. This is a typo. In the revised version we will keep a single notation and write $g_c = \phi(Z_c)$ everywhere so that the definition of the global context summary is consistent across the paper.
>
> By “global context summary” we mean a single vector that summarizes all contextual information available for one forecasting window across time and across channels. Intuitively this vector plays the role of a short description of the current regime of the series under its context, for example “high demand during a holiday with discount active” or “normal weekday traffic with no incident”. Formally, $Z_c$ collects the context token representations over time and channels for that sample and we apply a learnable summarization network $g_c = \phi(Z_c)$. In practice we first pool $Z_c$ along time and channels to obtain compact statistics such as means and maxima, then feed this pooled vector into a small MLP to produce $g_c$. This vector is then used to modulate all temporal queries in the ContextAdapt block so that the backbone reads its own history through the lens of the current context regime.
>
> We use $g_c$ rather than the raw tensor $Z_c$ or a plain average for both stability and expressiveness. Directly conditioning queries on the full $Z_c$ would make the modulation high dimensional and sensitive to local noise in individual channels and time steps. A sample level summary $g_c$ offers a more stable signal that is shared across all queries in that window. A simple mean $\bar Z_c$ would be more stable but it treats every context token as equally informative. The learnable map $\phi$ can instead learn to emphasize directions in $Z_c$ that are predictive for the target and to down weight directions that correspond to weakly related or noisy context. This is important for semantic selective query modulation because we want the backbone to bias its attention toward task relevant context semantics rather than an unweighted blend of everything. Finally, our context tensor mixes heterogeneous sources and may contain missing channels on some datasets. Since $\phi$ sees the masked $Z_c$ it can learn invariances to missing or unreliable channels and to scale differences across sources, which a fixed averaging rule cannot provide. We will clarify this definition and unify the notation for $g_c$ in the revised version.
>
>
>
> ---
>
> **Q2. On whether context is “equivalent to” a causal relationship.**
>
>
> Thank you for this question. We fully agree that context itself is not a causal relationship. Our goal is not to equate “context” with “causality”. Our goal is to design a context-aware mechanism that is guided by causal thinking and that encourages the model to rely on context signals that are stable drivers of the *future* target rather than on fragile correlations. In our conceptual view the underlying structure is that history and context jointly influence the future target. We write this informally as $(\text{history}, \text{context}) \rightarrow \text{future}$. The past trajectory is the main temporal driver of the future. Context contains exogenous or auxiliary variables that can shift or modulate this evolution.
>
> In the architecture we use the history representation as the query and the context representations as keys and values. This choice follows the standard cross-attention interface and is purely architectural. It does not mean that we assume that context causes history. Instead the ContextAdapt block learns how different context channels *modulate the mapping from history to future*. When we talk about “semantic selectivity between history and context” we mean that the model learns which context channels should be given influence when shaping the forecast that is based on history. That is, we are selecting which parts of the context are treated as plausible drivers or regime indicators for the future given the observed past.
>
> Our “causally inspired” components should be understood in this light. The channel wise importance vector and the counterfactual masking loss are inspired by ideas from causal reasoning such as focusing on stable drivers and checking importance through interventions. They do not perform formal causal identification and we do not claim to recover the true structural causal model of the system. What we enforce is that if the model marks a context channel as important then masking that channel should noticeably change its forecast. This aligns importance scores with behavior in an interventional sense but it is still an inductive bias inside a predictive model.

---

> ### Author Response · Authors · 2025-11-25
>
> **Q3 – How external covariates are defined and aligned with the history target**
>
> Thank you for asking for clarification here. In all experiments the external covariates are exactly the auxiliary variables that come with each benchmark. Depending on the dataset they include calendar indicators, weather measurements, mobility indices, Google Trends style signals, age specific incidence rates, and other domain specific exogenous variables released by the original authors. For each univariate target series we place these variables into a unified context matrix $C \in \mathbb{R}^{L \times N}$. Here $L$ is the history length used for forecasting and $N$ is the number of context channels. The $\ell$-th row of $C$ collects all covariate values that are defined at the same timestamp as the target value $t_\ell$. The common key for alignment is the time index used in the dataset, for example day or week or hour. For every time step in the target history we gather the covariates that are defined on that grid and write them into the corresponding row of $C$. Static or very slow moving attributes such as region demographics or group identifiers are broadcast along the time dimension of that series so that each time step carries the same static context.
>
> Missing or unavailable entries are recorded in a binary mask $M_c \in \{0,1\}^{L \times N}$. We multiply this mask with $C$ and then pass the masked matrix through a shared embedding module to obtain context tokens $Z_c$ as described in Eq. (2). This construction makes the model interface agnostic to the concrete type of covariate. Calendar flags, numeric sensors, and aggregated event features all become time aligned channels in the same matrix $C$. The ContextAdapt block only sees these channels and their mask, so the same mechanism can be applied across datasets with very different external signals. In the revised version we will add a short description and an illustrative figure in the appendix to make this alignment procedure more explicit.
>
>
>
>
> ---
>
> **Q4 – Why the counterfactual loss helps instead of hurting training**
>
> We agree that masking the most salient context channels can look risky at first sight. The key is that we never ask the model to solve the main forecasting task without these channels. During training we always run two coupled forward passes that share the same parameters. The factual pass uses the full context matrix and produces the prediction that is used in the primary forecasting loss. The counterfactual pass temporarily masks the Top K channels according to the current importance scores and produces a second prediction that is used only in the counterfactual loss term. The model is optimized so that the factual prediction is as accurate as possible, while the counterfactual prediction is allowed to be worse as long as this difference is consistent with the learned importance scores.
>
> The counterfactual loss is therefore a regularizer that shapes how the model assigns and uses importance, not a replacement objective for forecasting. Intuitively it enforces the following behavior. If a channel is marked as highly important, then removing it in the counterfactual pass should noticeably increase the error. If removing a channel barely changes the error, then the gradients from the counterfactual term push its importance score down. Because both passes share all weights, this regularizer adjusts both the channel score vector and the forecasting network itself. Over training the model settles on a configuration where the channels that it truly relies on for prediction are exactly those that receive high importance scores, and channels that are redundant or only weakly correlated do not get promoted.
>
> This alignment between declared importance and behavioral impact is what makes the loss helpful. It discourages the model from using attention or channel scores as a purely cosmetic explanation layer that does not reflect how the forecast is actually produced. At the same time the factual path remains the anchor for predictive performance since its loss term is dominant and it always sees the full context. At inference time we only use this factual path. The masked counterfactual branch is never used to generate predictions. In this way the counterfactual loss sharpens and validates context usage during training, while the model at test time still fully exploits the most salient context channels.
>
> **Q5. Typo for “Semantic Slectivity”**
>
> Thank you for catching this. It should read “Semantic Selectivity”. We will fix this typo and thoroughly re-check all figures and captions in the revised version to avoid similar issues.

---

### Meta-Review · Area_Chair_FZ8z · 2026-01-06

**Summary:**

This paper has been assessed by four knowledgeable reviewers. Three of them recommended rejecting it (all marginally) while the fourth gave it a marginal acceptance score. The paper proposes a new forecasting approach that integrates internal and external context, shows good performance on varied datasets, and gives sensible intuition about selecting external features. However, the reviewers noted that its methodological novelty is limited, and the work repeatedly frames itself in causal terms without providing actual causal inference, identifiability guarantees, or validation of causal claims—reducing clarity and potentially misrepresenting the contribution. The method appears to exacerbate computational costs, lacks analysis under distribution shift, and would benefit from clearer theoretical grounding, simpler baselines, and more rigorous evaluation of the proposed feature‑selection mechanism. The authors provided comprehensive replies to most of the concerns raised by the reviewers, but overall this paper is slightly below the ICLR acceptance threshold.

**Reviewer Concerns:**

Some of the concerns appear addressable and the authors made a good attempt at doing that. However, e.g., even if the excessive causality claims can be softened so that the final revision of the paper correctly reflects the merits of the proposed method, such change will also reduce the face value of the proposed method.

**Reviewer Scores:**

I do not feel that the rebuttal would compel the reviewers to raise their scores enough for this paper to cross the acceptance threshold at this time.

---

### Decision · Program_Chairs · 2026-01-26

Reject